# Wavenumber-dependent transmission of subthreshold waves on electrical synapses network model of *Caenorhabditis elegans*

**Iksoo Chang[1,2,3], Taegon Chung[1], Sangyeol Kim[1,2]***

[1]Department of Brain Sciences, Daegu Gyeongbuk Institute of Science and Technology, Daegu, Republic of Korea; [2]Creative Research Initiative Center for Proteome Biophysics, Daegu Gyeongbuk Institute of Science and Technology, Daegu, Republic of Korea; [3]Supercomputing Bigdata Center, Daegu Gyeongbuk Institute of Science and Technology, Daegu, Republic of Korea

## eLife Assessment

This study presents numerical results on a framework for understanding the dynamics of subthreshold waves in a network of electrical synapses modeled on the connectome data of the C elegans nematode. The strength of the evidence presented in favor of interference effects being a major component in subthreshold wave dynamics is **inadequate** and the approach is flawed. Substantial methodological issues are present, including altering the original network structure of the connectome without a clear justification and providing little motivation for the choice of numerical parameters values that were used.

**\*For correspondence:**
sykim@dgist.ac.kr

**Competing interest:** The authors declare that no competing interests exist.

**Abstract** Recent experimental studies showed that electrically coupled neural networks like in mammalian inferior olive nucleus generate synchronized rhythmic activity by the subthreshold sinusoidal-like oscillations of the membrane voltage. Understanding the basic mechanism and its implication of such phenomena in the nervous system bears fundamental importance and requires preemptively the connectome information of a given nervous system. Inspired by these necessities of developing a theoretical and computational model to this end and, however, in the absence of connectome information for the inferior olive nucleus, here we investigated interference phenomena of the subthreshold oscillations in the reference system *Caenorhabditis elegans* for which the structural anatomical connectome was completely known recently. We evaluated how strongly the sinusoidal wave was transmitted between arbitrary two cells in the model network. The region of cell-pairs that are good at transmitting waves changed according to the wavenumber of the wave, for which we named a wavenumber-dependent transmission map. Also, we unraveled that (1) the transmission of all cell-pairs disappeared beyond a threshold wavenumber, (2) long distance and regular patterned transmission existed in the body-wall muscles part of the model network, and (3) major hub cell-pairs of the transmission were identified for many wavenumber conditions. A theoretical and computational model presented in this study provided fundamental insight for understanding how the multi-path constructive/destructive interference of the subthreshold oscillations propagating on electrically coupled neural networks could generate wavenumber-dependent synchronized rhythmic activity.

## Introduction

The ion current flowing through the gap junction between neurons changes the membrane potential of those neurons, which is called signal transmission by electrical synapses (*Carew and Kandel, 1976*). Electrical synapses have characteristics such as simple structure, fast signal transmission, bidirectional communication (in most cases), and a wide operation range for voltage including the subthreshold region (*Bennett, 1997*). Based on this, it has been reported that electrical synapses play various roles, such as excitation or inhibition of neurons, synchronization or desynchronization of neural activity, promotion or attenuation of rhythms, and improvement of signal-to-noise ratio (*Connors, 2017*). Here, rhythm refers to the periodic active pattern of neurons, and in particular, the rhythm in the subthreshold region helps neurons ignite the action potential simultaneously according to the cycle of the rhythm, and a typical example of this can be seen is the excitatory cells in inferior olivary nucleus (*Bazzigaluppi et al., 2012*; *Benardo and Foster, 1986*; *Giocomo et al., 2007*; *Khosrovani et al., 2007*; *Long et al., 2002*). Spontaneous and robust subthreshold membrane potential oscillations were observed in the inferior olivary cells (*Benardo and Foster, 1986*; *Chorev et al., 2007*; *Devor and Yarom, 2002*; *Lampl and Yarom, 1997*; *Llinás and Yarom, 1981*). The inferior olivary cells are intra-connected only by electrical synapses, and it was reported that the presence of electrical synapses is largely involved in the synchronization of subthreshold oscillations (*De Zeeuw et al., 2003*; *Leznik and Llinás, 2005*; *Leznik et al., 2002*; *Llinás, 2013*; *Long et al., 2002*; *Turecek et al., 2016*). Although electrical synapses and subthreshold oscillations are functionally closely related, theoretical and computational exploration of this has been lacking. The occurrence of subthreshold oscillations and their role in neural networks have been explored through pioneering neuron models (*Roach et al., 2018*; *Tchumatchenko and Clopath, 2014*; *V-Ghaffari et al., 2016*), but the establishment of an entire connectome between numerous neurons belonging to regions such as the inferior olivary nucleus remains a task to be solved in the future.

In the field of connectome, information on the indexing of all individual neurons, muscles, and organ cells present in the nematode *Caenorhabditis elegans* and the entire connectivity between those cells by chemical and electrical synapses was recently updated (*Cook et al., 2019*). This available connectome data of *C. elegans* identifies connectivity weights proportional to the structural size of each synapse, with 1433 electrical synapses present among a total of 469 cells, including neurons, muscles, pharynx, and other organ cells. Some examples of synchronized activity of a small number of neurons through electrical synapses in *C. elegans* have been reported. During the gentle nose touch response, nose mechanoreceptor neurons connected to electrical synapses simultaneously increased in activity, thereby serving as a coincidence detector (*Chatzigeorgiou and Schafer, 2011*). And a pair of GABAergic motor neurons controlling the bowel movement program repeated every 50 s used synchronized activity through electrical synapses at their axon terminal (*Choi et al., 2021*). However, direct observation or indirect evidence of spontaneous subthreshold oscillations has NOT been confirmed in *C. elegans*. Nevertheless, a theoretical and computational model consideration of the signal transmission of subthreshold oscillations on the entire electrical synapse network among cells with various cell types could be a great challenge.

In general, the wave properties of membrane potential are well-known, such as constructive summation of the two excitatory postsynaptic potentials or deconstructive summation of the excitatory and inhibitory postsynaptic potentials. Likewise, it is experimentally confirmed that subthreshold oscillations have wave properties as fluctuations observed in membrane potential measurements and experience interference when propagated in the neural tissue space (*Chiang and Durand, 2023*; *Gupta et al., 2016*). Therefore, interference should be considered important when subthreshold membrane potential waves propagate through the neural network and cross each other at one spatial point.

Here, we attempted to describe a situation in which wave signals, mimicking the subthreshold membrane potential waves, are propagated and interfered on a network that mimics the electrical synapse network of *C. elegans* (the cells become nodes, the electrical synapses become edges, and the subthreshold waves can flow in both directions on the edges). In the field of physics, the Anderson localization phenomenon has been well known, and it is a general wave phenomenon due to the wave interference between multiple scattering paths. Depending on the strength of the scattering of waves, the result of wave interference could be constructive or destructive, namely, waves could transmit or localize through the media, correspondingly. The theoretical and numerical method for describing the so-called Anderson localization problem usually employed the tight-binding Anderson

Hamiltonian. A relevant model was developed to calculate the wave's quantum percolation using the tight-binding Anderson Hamiltonian on a network lattice consisting of nodes and edges, where the interference by all possible propagation paths was considered when the quantum mechanical probability wave was propagated by experiencing the percolation disorder on the network lattice (*Anderson, 1958*; *Chang et al., 1995*; *Meir et al., 1989*; *Shapir et al., 1982*; *Thomas and Nakanishi, 2016*). In this study, we benefited from this model, and by treating the probability wave as a general wave signal and replacing the network lattice with a synapse network model we are interested in. We could calculate the interference by all possible paths which were considered when the wave signals propagate in the synapse network.

By applying the theoretical concept and computational methodology introduced above and well established in the field of physics, we calculated the signal transmission coefficient between arbitrary two cells of the model connectome as a function of the wavenumber (the quantity of inverse of wavelength) of the wave signal. Based on the signal transmission coefficients for all cell-pairs estimated under various wavenumber conditions, we unraveled (1) the existence of a threshold wavenumber above which the signal transmission disappears, (2) regions of cell-pairs with wavenumber-dependent transmission map, (3) long distance and regular patterned signal transmission, and (4) major hub cell-pair common across multiple wavenumbers. When comparing with the results for a situation that did not consider the wave interference, we revealed the role of the interference when subthreshold waves propagate on the *C. elegans*' electrical synapse network model. Although many aspects of the actual neural network and electrical synapses were simplified or omitted in our model study, our results provided the insight of the fundamental role of the electrical synapse network with subthreshold oscillations.

## Results

### Cell-pairs in the model connectome of *C. elegans* could belong to regions where the wave signal could be transmitted (or not transmitted) below (or above) a threshold wavenumber

In our circuit model mimicking the anatomical gap junction network of *C. elegans* (details described in 'Construction of our circuit model' part of Methods) (*Figure 1*), we calculated all the transmission coefficient $T_{ij}$ of the wave signal for each of the 109,746 cell-pairs $\langle i, j \rangle$ between the total 469 cells (details described in 'Calculation for the transmission coefficient' part of Methods). Here, the transmission coefficient was a function of energy $E$ of the incoming wave, and the $E$ value was a function of the wavenumber $k$ of the wave signal. And according to the wavenumber, $E$ value was set up to exist in the range [–10, 10], and we calculated all the $T_{ij}(E)$ of the entire cell-pairs at $E$ values of 801 points listed by equal intervals over the entire range ($\Delta E = 0.025$).

First, the average transmission coefficient of all cell-pairs $\langle i, j \rangle$ for each $E$ value, $\langle T_{ij}(E) \rangle_{All\ ij\ cell-pairs}$, was obtained and drawn as a function of $E$ (*Figure 2A*). The $\langle T(E) \rangle$ graph showed an incomplete but quite symmetrical form based on $E = 0$. And on the real $Y$-axis scale, high $\langle T(E) \rangle$ values were broadly identified in these three ranges of the $E$ values: [–1.575, –1.275], [–0.350, 0.350], and [1.275, 1.575]. Also on the log $Y$-axis scale, several relatively high $\langle T(E) \rangle$ values were locally identified in the form of peaks in the remaining ranges of the $E$ values. Meanwhile, on the log $Y$-axis scale, the signal transmission of all cell-pairs almost disappeared in the $E$ value range at both ends. In the field of solid-state physics, a threshold energy value in which electrical conductivity disappears in the energy band is called a 'mobility edge'. We were able to define a threshold wavenumber, namely a signal mobility edge, even in the transmission of the wave signal of our circuit model. For example, considering that $\langle T(E) \rangle = 10^{-7}$ is a value that can be obtained when only one cell-pair out of all cell-pairs shows 1% transmission and all others are completely unable to transmit the wave, this $10^{-7}$ value can be regarded as an extreme disappearance of signal transmission throughout the entire circuit. Thus, by taking this reference value, $E = \pm 5.650$ in our circuit model became a signal mobility edge. If the reference value is set more conservatively lower than now, the absolute value of $E$ corresponding to the signal mobility edges will be larger than now.

Next, the transmission coefficient of all individual cell-pairs $T_{ij}(E = 0.225)$ was represented as a heatmap at the $E$ value (or the corresponding wavenumber) where the $\langle T(E = 0.225) \rangle$ value was the highest (*Figure 2B*). Here, we confirmed that not all cell-pairs transmit the wave signals with even

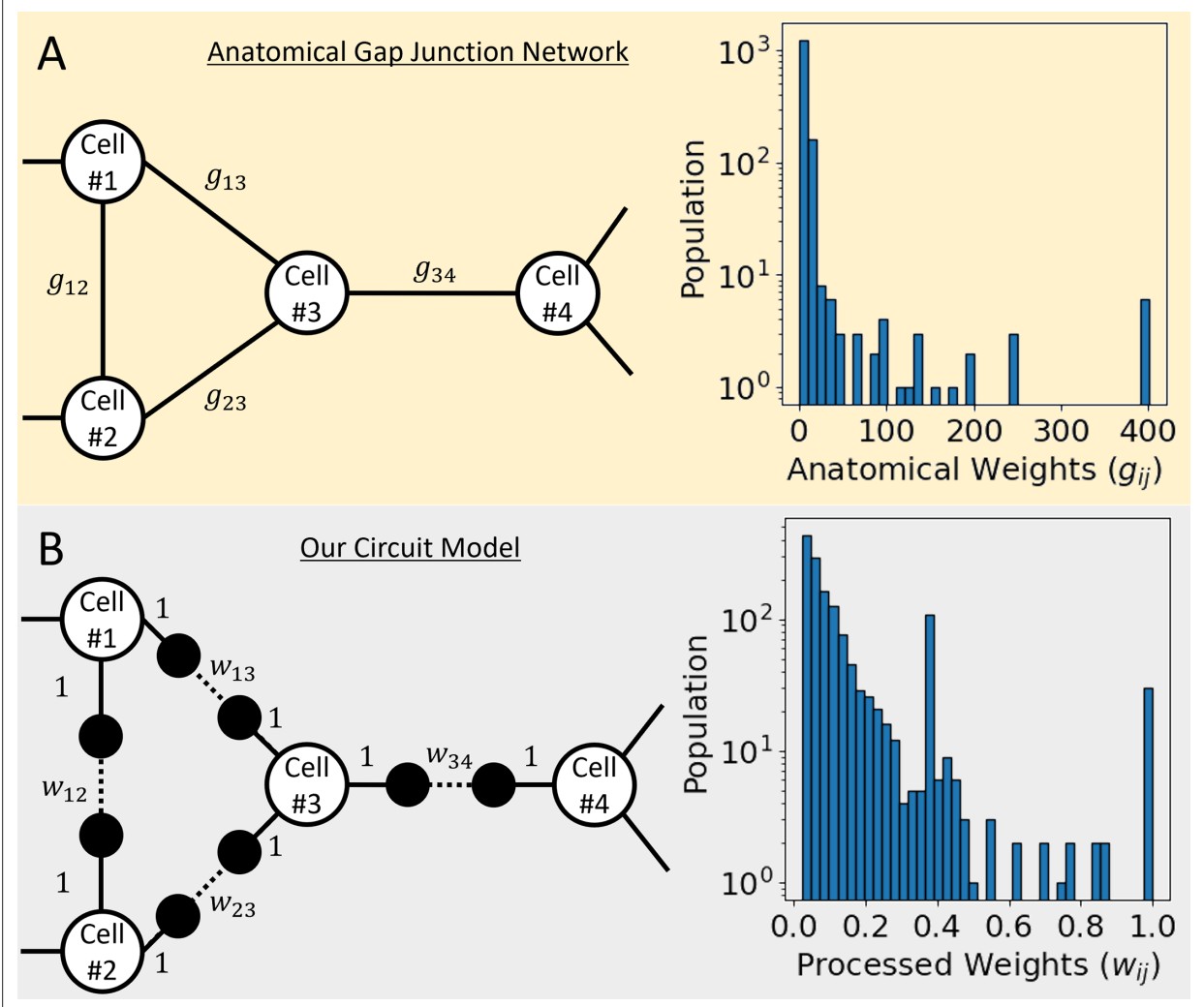

**Figure 1.** Anatomical gap junction network and our circuit model. (**A**) In the sample of the network diagram on the left, there are white-filled nodes stand for cells, and the edges of a solid line stand for the anatomical gap junction between them. The graph on the right is the distribution of 1433 gap junction weights obtained from the connectome data of *C. elegans*. (**B**) In the sample of the network diagram on the left, equally spaced two virtual nodes were added between cells connected by a gap junction, and the virtual node is presented as a black-filled circle. The solid line stands for the connection between the cell and the virtual node that gave a weight of 1, and the dotted line stands for the connection between the two virtual nodes that gave our processed weight between 0 and 1. Our processed weights are calculated from the anatomical gap junction weights, and the graph on the right shows its distribution.

transmission coefficients. Most of the strong transmissions were found in the cell-pairs of intra-body-wall muscles, and a small number of cells belonging to sensory neurons, motor neurons, and other organ cells also showed strong transmission in cell-pairs between them or to body-wall muscles. The rest of the cells except these cells were almost isolated separately without exchanging the wave signals with any cells even at this $E$ value, which was the highest transmitted state on average.

In addition, the transmission coefficient of all individual cell-pairs $T_{ij}(E = 5.650)$ was represented as a heatmap at the $E$ value (or the corresponding wavenumber) which we defined as the signal mobility edge (*Figure 2C*). As expected, all cells were isolated separately without exchanging the wave signals with each other, and the entire circuit stop transmitting wave signals of this wavenumber.

## Transmission map of subthreshold waves on the electrical synapse network model depended on values of wavenumber

Between the two $E$ values discussed above ($E = 0.225$ and $E = 5.650$), the average transmission coefficient of all cell-pairs, $\langle T(E) \rangle$, did not simply increase or decrease, but fluctuated along the $E$ value

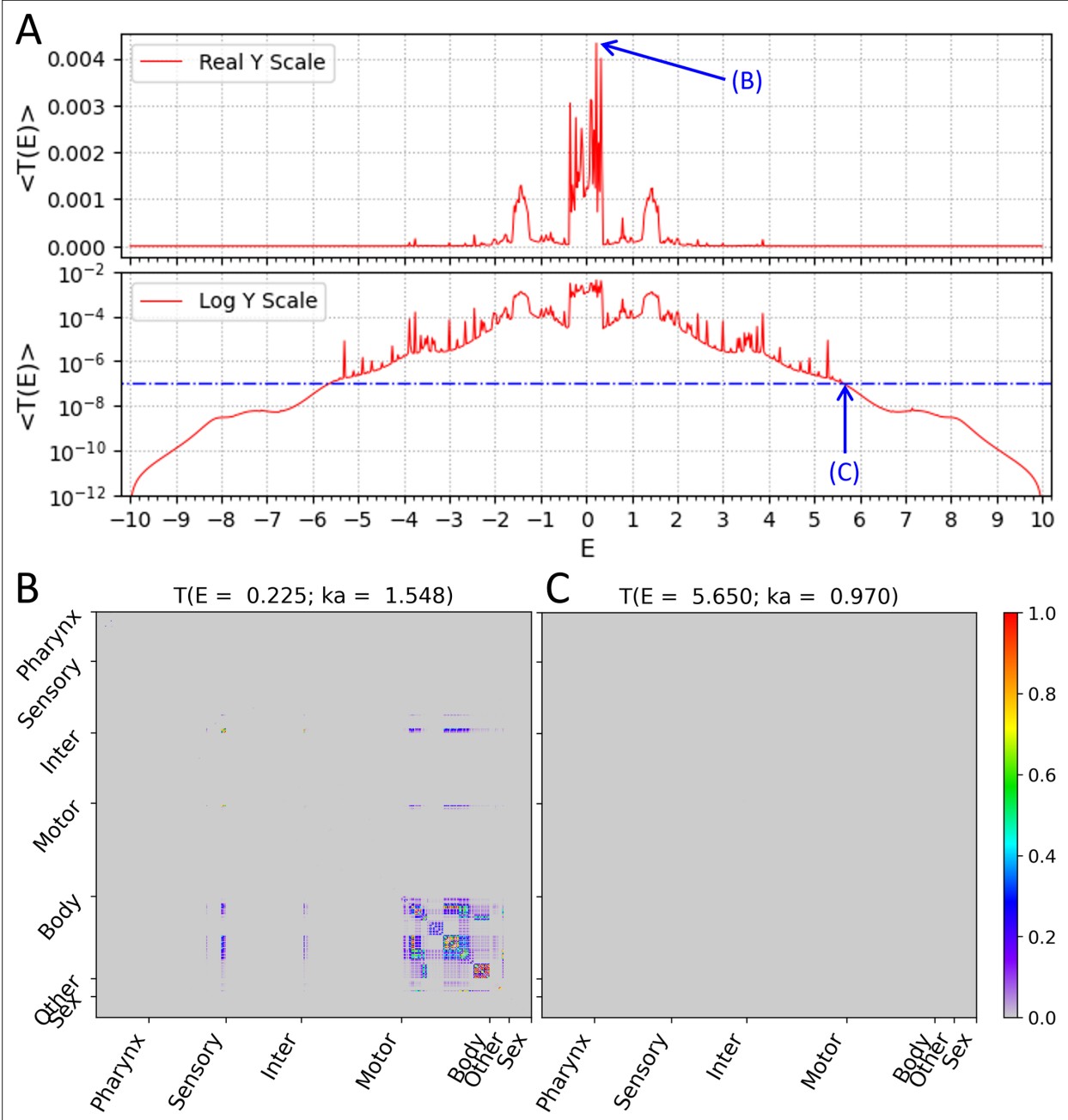

**Figure 2.** Wavenumber-dependent transmission coefficient. Average transmission coefficient for all cell-pairs as a function of $E$ values ($E$ value is a function of the wavenumber of the subthreshold wave). (**A**) The upper panel shows the average transmission coefficient graph on the $Y$-axis of the real scale and the lower panel shows the same graph on the $Y$-axis of the log scale. The reference value ($10^{-7}$) of the signal mobility edge is indicated in the lower panel as a dot-dash line. (**B**) The transmission coefficient of all cell-pairs at $E = 0.225$ is represented as a heatmap, which is the highest average transmission coefficient in our result. (**C**) The transmission coefficient of all cell-pairs at $E = 5.650$ is represented as a heatmap, which is one of the signal mobility edges. (**B, C**) In heatmap, the $X$ and $Y$ axes are arranged according to the index of 469 cells, and the index follows that of the original connectome data. Instead of displaying all cell names, only seven cell types were indicated.

The online version of this article includes the following figure supplement(s) for figure 2:

**Figure supplement 1.** Selected $\langle T(E) \rangle$ peaks.

(*Figure 2*). Therefore, it was expected that the composition of the cell-pairs, which showed strong transmission, also changes according to the $E$ value. To efficiently investigate the changes in the composition of the cell-pairs, instead of inspecting the $T_{ij}(E)$ heatmaps in all $E$ values, we selected 31 specific $E$ values and tried to analyze the composition of the cell-pairs only in these $E$ values. We

chose the specific $E$ values on the following three criteria: (1) the $\langle T(E) \rangle$ value was the local maximum (the shape of a peak) in the $\langle T(E) \rangle$ graph, (2) the components of the $T_{ij}(E)$ showed greater than 0.5 at least one cell-pair, and (3) the $E$ value was positive number (*Figure 2—figure supplement 1*). The reasons why we used these conditions are as follows: (1) we assumed that the peak occurred in the $\langle T(E) \rangle$ graph due to the emergence of new cell-pairs with strong transmission, (2) we set the reference value for strong transmission to be 0.5 or higher, and (3) the $E$ values where peak occurred in the $\langle T(E) \rangle$ graph showed left–right symmetry based on $E = 0$. (Additional explanations are in 'Auxiliary paragraph 1' of Supplementary Materials.)

The transmission coefficients of all individual cell-pairs were represented as a heatmap at each $E$ value (or the corresponding wavenumber) for the 31 specific $E$ values we selected (left panels in *Figure 3*, *Figure 3—figure supplements 1–8*). To closely observe the composition of cell-pairs with strong transmission, only cells (*Appendix 1—tables 1–4*) belonging to the cell-pairs with $T_{ij}(E) > 0.5$ were selectively exhibited in these heatmaps, not for all 469 cells. To confirm the spatial status of the cell-pairs with strong transmission on the electrical synapse network model, a network diagram was prepared first. Since drawing all 469 cells as a network diagram was complicated and poorly visible, we presented nodes for only 173 cells that belonged to the cell-pairs with $T_{ij}(E) > 0.5$ in the positive $E$ values. Electrical synapses between cells were represented as edges, and the thickness of the edge proportional to our processed weights ($w_{ij}$) was used. (Additional explanations are in 'Auxiliary paragraph 2' of Supplementary Materials). On the prepared network diagram, we marked the cell-pairs with strong transmission that satisfies $T_{ij} > 0.5$ (right panels in *Figure 3*, *Figure 3—figure supplements 1–8*).

As expected above, the region of the cell-pairs with strong transmission changed depending on the value of $E$ (or wavenumber), which we named 'Wavenumber-Dependent Transmission Map (WDTM) of subthreshold waves on the electrical synapse network model'. However, it should be also noted that our WDTM was a result derived from a network where only electrical synapses were considered, not a general neural network where chemical and electrical synapses coexist.

We looked at the cell type of the cell-pairs that make up the 31 WDTMs (*Figure 3*, *Figure 3—figure supplements 1–8*). In the two $E$ value ranges of [0, 0.350] and [1.275, 1.575], the high $\langle T(E) \rangle$ value was broadly identified (*Figure 2—figure supplement 1*), and many cell-pairs of intra-body-wall muscles were observed in the corresponding WDTMs of these $E$ values (*Figure 3A, C*, *Figure 3—figure supplements 1 and 2A, B*, and *Figure 3—figure supplement 4*). Since the body-wall muscles consist of four muscle strands: dorsal/ventral-left/right strands, the cell-pair of intra-body-wall muscles can be divided into two cases: a cell-pair of intra-strand and a cell-pair of inter-strand. Following this classification, the cell-pairs of both intra- and inter-strand were found in the WDTMs of the $E$ value range of [0, 0.350] (*Figure 3A*, *Figure 3—figure supplements 1 and 2A, B*), whereas only the cell-pairs of intra-strand were found in the WDTMs in the $E$ value range of [1.275, 1.575] (*Figure 3C* and *Figure 3—figure supplement 4*). On the other hand, in the remaining WDTMs, a small number of cell-pairs within cells with less than 10 members were mostly found (*Figure 3B, D*, *Figure 3—figure supplement 2C, D*, *Figure 3—figure supplement 3*, and *Figure 3—figure supplements 5–8*), and the WDTMs of the two largest cases are shown in *Figure 3B, D*. The composition of cell-pairs with various combinations of cell types (sensory/inter/motor neurons, body-wall/sex-specific muscles, pharynx, and other organ cells) was shown in these remaining WDTMs.

## Cell-pairs in body-wall muscles on long distance with regular patterned transmission were found in the WDTMs

When looking at each WDTM on the network diagram, it shows various patterns spatially. In particular, in the case of the body-wall muscles, the cell-pairs of intra-strand existed, ranging from short distance to the longer distance than the half of the full length of the strand (*Figure 3A, C*). Here, the 'Distance' was meant by the length of the shortest path along the network between two cells of a cell-pair. If two cells are directly connected to an edge, the distance between the two cells is one. In the WDTMs of the $E$ value range of [1.275, 1.575], cell-pairs with regular patterned transmission of the body-wall muscles were observed through the network diagram (*Figure 3—figure supplement 4*). Only the cell-pairs of intra-strand were found in these WDTMs, and the configuration of the distance of these cell-pairs was different for each WDTM (multiple of 3 at $E = 1.350$, multiple of 2 at $E = 1.450$, multiple of 7 at $E = 1.500$, and multiple of 4 at $E = 1.575$).

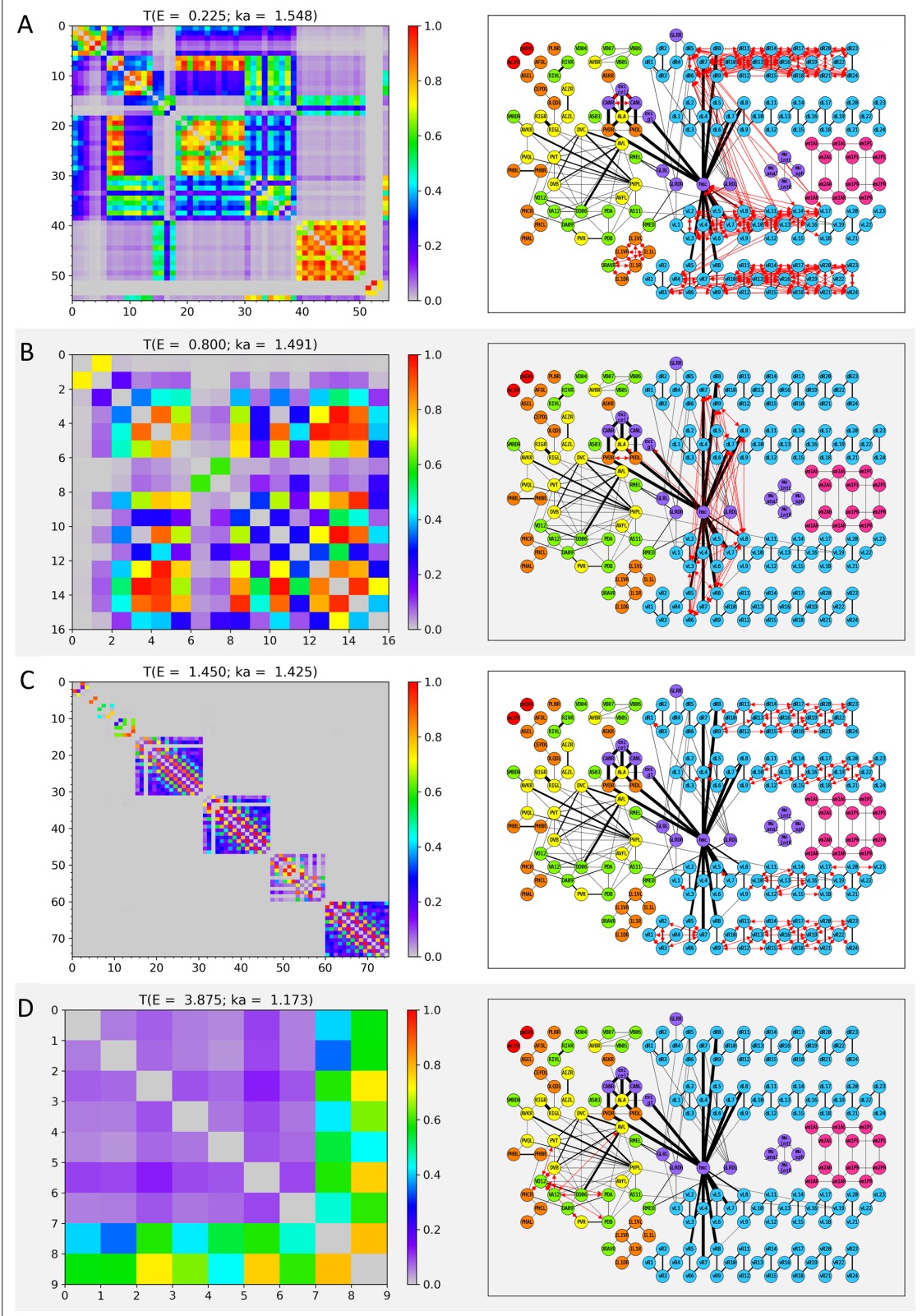

**Figure 3.** Wavenumber-dependent transmission map. The cell-pairs with strong transmission ($T_{ij}(E) > 0.5$) at (**A**) $E = 0.225$, (**B**) $E = 0.800$, (**C**) $E = 1.450$, and (**D**) $E = 3.875$ (or the corresponding wavenumbers) are shown, respectively. The left panels represent the transmission coefficient between cells belonging to the cell-pairs with strong transmission at each $E$ value as a heatmap. Here, the $X$ and $Y$ axes are the cell index, and the cell name corresponding to the cell index can be found in *Appendix 1—table 1*, *Appendix 1—table 2*, *Appendix 1—table 3*, *Appendix 1—table 4*.

*Figure 3 continued on next page*

*Figure 3 continued*

The right panels display the cell-pairs of strong transmission at each *E* value as red bidirectional arrows of the same thickness on the network diagram. In the network diagram, the cell name is written inside the circle that stands for each node (for the body-wall muscles, the cell name is abbreviated as follows: ex) dBWML1 → dL1, and the color of the node stands for the cell type (red: pharynx cells, orange: sensory neurons, yellow: inter neurons, green: motor neurons, light blue: body-wall muscles, purple: other end organs, magenta: sex-specific cells). Edges indicated by black solid lines stand for the electrical synapses among the cells, and the thickness of the edges is proportional to our processed weight. In the network diagram, only 173 cells that belonged to the cell-pair with strong transmission at least once in positive *E* values are represented, and the remaining cells and the electrical synapses by them are omitted from display. The virtual nodes of our circuit are also omitted from the display.

The online version of this article includes the following figure supplement(s) for figure 3:

**Figure supplement 1.** Wavenumber-dependent transmission map.

**Figure supplement 2.** Wavenumber-dependent transmission map.

**Figure supplement 3.** Wavenumber-dependent transmission map.

**Figure supplement 4.** Wavenumber-dependent transmission map.

**Figure supplement 5.** Wavenumber-dependent transmission map.

**Figure supplement 6.** Wavenumber-dependent transmission map.

**Figure supplement 7.** Wavenumber-dependent transmission map.

**Figure supplement 8.** Wavenumber-dependent transmission map.

On the other hand, when the cell-pairs of inter-strand of the body-wall muscles appeared in the WDTMs as shown in *Figure 3A, B*, the head mesodermal cell (hmc) seemed to play an important role in building a bridge between them. This would be recognized from the network diagram itself of black edges without WDTM, and hmc was strongly connected near the neck of the body-wall muscles (seventh and eighth muscles) for all four strands. In the WDTM of *Figure 3A*, there were many cell-pairs of inter-strand between dorsal-right and ventral-left strands, and there were also many cell-pairs between the ventral-left strand and hmc, which directly revealed the role of hmc as a bridge between the two body-wall muscle strands. In the WDTM of *Figure 3B*, it was shown that the role of hmc was indirectly revealed as the cell-pairs of inter-strand, which existed only among the neck of four body-wall muscle strands.

Additionally, we reaffirmed the important role of hmc while confirming the distance characteristics of the cell-pairs with strong transmission. Under the conditions used when preparing the network diagram (the transmission coefficient $T_{ij}(E)$ of a cell-pair was 0.5 or higher in the range of positive *E* values at least once), a total of 146 cell-pairs with strong transmission were identified excluding the cell-pairs of intra-body-wall muscles, and their distance was investigated (*Appendix 1—table 5*). (Additional explanations are in 'Auxiliary paragraph 3' of Supplementary Materials.) Most of the cell-pairs in *Appendix 1—table 5* showed the kinds of short distances 1–3, which consisted of various cell types. On the other hand, the cell-pairs with long distances in *Appendix 1—table 5* ranging from 4 to 17 were mainly pairs between hmc and body-wall muscles. The strong transmission between hmc and the tail muscle, which corresponds to the longest distance, passes through the intra-strand connections from the neck to the tail of body-wall muscles.

A few cell-pairs of strong transmission with long distance in *Appendix 1—table 5* turned out to be not those between hmc and the body-wall muscles. In there, one inter neuron (AVBR) and two motor neurons (AS11 and VD04) formed a cell-pair with the body-wall muscles, respectively, and four sensory neurons (CEPDL, IL1R, IL1DR, and OLQDL) formed cell-pairs between them.

## Major hub cell-pairs with the strong transmission of signals for many wavenumbers exist

We looked at cell-pairs frequently appearing in all 31 WDTMs in order to identify the major hub cell-pairs that is commonly used for various wavenumbers. First, for each cell-pair $\langle i, j \rangle$, we calculated the average appearance rate in all 31 WDTMs, $\langle \theta [T_{ij}(E) - 0.5] \rangle_{E \in \{31 \text{ selected } E\}}$, and represented this as a heatmap (*Figure 4*). Where, the $\theta[x]$ was a step function, which value was 1 if $x \geq 0$ or 0 if $x < 0$. Thus, the average appearance rate was 1 in the case of a cell-pair that appeared in all 31 WDTMs, and $1/31 \approx 0.03$ in the case of a cell-pair that appeared only once. In this study, we declared a cell-pair as a major hub cell-pair if the average appearance rate of a cell-pair was $x < 0$ or higher. The major

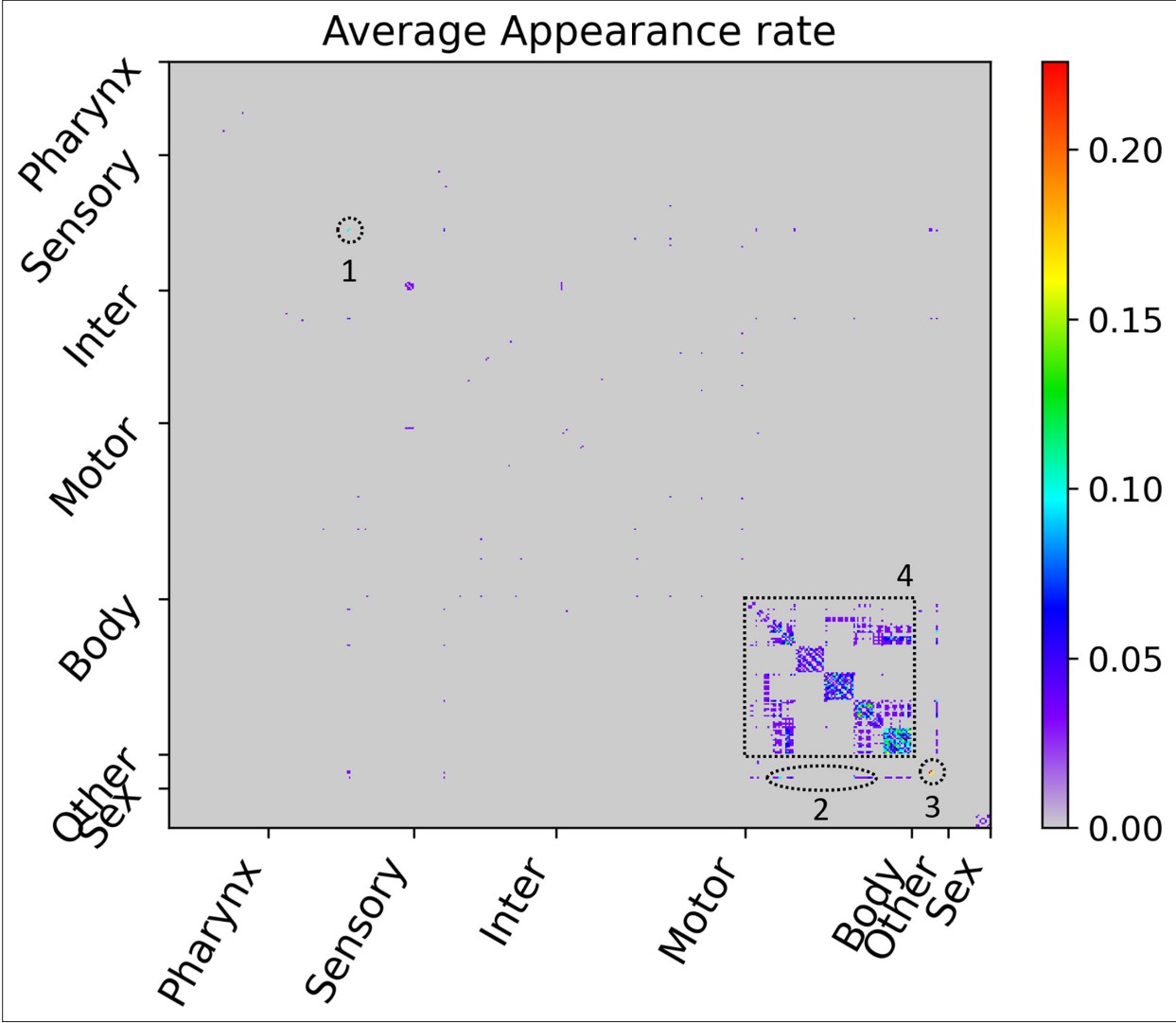

**Figure 4.** Average appearance rate as cell-pair with strong transmission in the WDTMs. The average appearance rate of all cell-pairs is represented as a heatmap. Here, the X and Y axes are arranged according to the index of 469 cells, and the index follows that of the original connectome data. Instead of displaying all cell names, only seven cell types were indicated. The highest rate is $7/31 \approx 0.23$, and four cell-pair groups showing rates of $3/31 \approx 0.1$ or higher are identified and numbered in the heatmap. (1) PVDL-PVDR pair, (2) hmc-vBWML6, hmc-vBWML7, and hmc-vBWML8 pairs, (3) CANL-CANR, CANL-exc_cell, and CANR-exc_cell pairs, and (4) many cell-pairs of intra-strand of the body-wall muscles.

hub cell-pairs were the cell-pairs of intra-strand of the body-wall muscles (*Figure 4*). As seen in the network diagram, the edges of each strand of the body-wall muscles from the neck (seventh or eighth muscle) to the tail (twenty-third or twenty-fourth muscle) were simply connected in a line, unlike the messy edges among most neurons in the network (*Figure 3*). In general, the complexity of connections among most neurons increase with the number of transmissible paths of wave propagation, and the sum of phase changes coming from various path-length differences causes the deconstructive interference. Therefore, the simple (or regular) connections in each strand of the body-wall muscles from the neck to the tail reduced the number of transmissible paths of wave signals, which suppressed deconstructive interference and thus resulted in the strong transmission for various wavenumbers.

Also, the cell-pairs between hmc and the neck of the ventral-left strand of the body-wall muscles were also identified as major hub cell-pair (*Figure 4*). As seen in the network diagram, since the edges between hmc and the neck muscles were very strong compared to other edges (*Figure 3*), the relative contribution of wave signals coming from other cells could be almost ignored, and thus resulted in the strong transmission for various wavenumbers.

The cell-pairs among two sensory neurons (PVDL and PVDR) and three other organ cells (CANL, CANR, exc_cell) were also identified as major hub cell-pairs (*Figure 4*). The cell-pair between CANL and CANR was the most major hub cell-pair, with the highest average appearance rate of $7/31 \approx 0.23$. As seen in the network diagram, the three other organ cells were very strongly connected by the edges to each other, and the two sensory neurons were very strongly connected by the edges to both ALA inter neuron and hmc (*Figure 3*). Although there were cases where they were strongly connected by the edges to each other, such as between PVDL and CANL or between PVDR and CANR, they were not major hub cell-pair. Therefore, the strong edge shown on the network diagram was not a sufficient condition to implicate a major hub cell-pair.

## Discussion

The fundamental aspect of this study is that the interference phenomena of the subthreshold oscillations propagating on a neuronal circuit was described by a theoretical and computational model study of the reference system *C. elegans* for which the structural anatomical connectome was completely known. We investigated the wavenumber-dependent transmission of the sinusoidal wave signals between all cell-pairs on the connectome model of *C. elegans* by considering the interference effect caused by multi-paths of signaling.

The results related to the wavenumber-dependent transmission map expanded the way we used to view signal transmission on electrical synapses. The difference in signal transmission with the presence or absence of interference effect on electrical synapses could be illustrated by considering the effective conductance between two nodes in the electric circuit with or without considering the interference effect (*Figure 5*). For a situation without considering the interference, we calculated the effective resistance and effective conductance for all cell-pairs by imitating a cell as a node and the electrical synapse as an electrical resistor connecting nodes as shown in *Figure 5A* (details in 'Calculation for effective conductance' part of Methods). Several inter neurons and other organ cells showed high effective conductance, while pharynx, body-wall muscles from the neck to the tail, and sex-specific muscles showed very low effective conductance (*Figure 5B*). Since inter neurons have many complex connections among them, many electrical current paths exist between the two inter neurons, which act as parallel connections of multiple electrical resistors. This results in lower effective resistance (or higher effective conductance). However, for a situation with considering the interference, many transmissible forward and backward paths of the signal propagation rather caused the deconstructive interference, preventing the transmission of the signal. For each cell-pair $\langle i, j \rangle$, we calculated the average transmission coefficient for all $E$ values and represented this as a heatmap (*Figure 5C, D*). Which clearly showed that most of the neurons could not exchange wave signals with each other except for very few neurons. Wave signals were effectively delivered only within a small number of very strongly connected cell groups (PVDL/R, CANL/R, exc_cell, and hmc) or within a large number of very regularly connected cell groups (body-wall muscle strands from the neck to the tail).

In this study, we used the structural anatomical information from *C. elegans*' electrical synapse network, but no synchronized rhythmic activity induced by subthreshold membrane potential oscillations has been reported for *C. elegans* so far. However, we aimed at investigating the signaling characteristics caused by the interference and to discover a feasible phenomenon related to the propagation of the subthreshold oscillations on a neuronal circuit of living nervous system. As to the synchronized rhythmic activation observed like in the mammalian inferior olive nucleus, we think that there might be an electrical synapse network in there that connect cells very strongly or very regularly. The plausible possibility according to our model study is that the constructive interference of subthreshold membrane potential waves with a specific wavenumber may generate the synchronized rhythmic activation. We hope that the results in our study would serve as the worthwhile framework and knowledge for designing the future experimental studies not only for inferior olive nucleus, *C. elegans* but also other living systems.

## Methods
### Construction of our circuit model
We used the contents of the 'hermaphrodite gap jn symmetric' tab in the 'SI 5 Connectome adjacency matrices, corrected July 2020.xlsx' file of the *C. elegans* connectome dataset as data for the

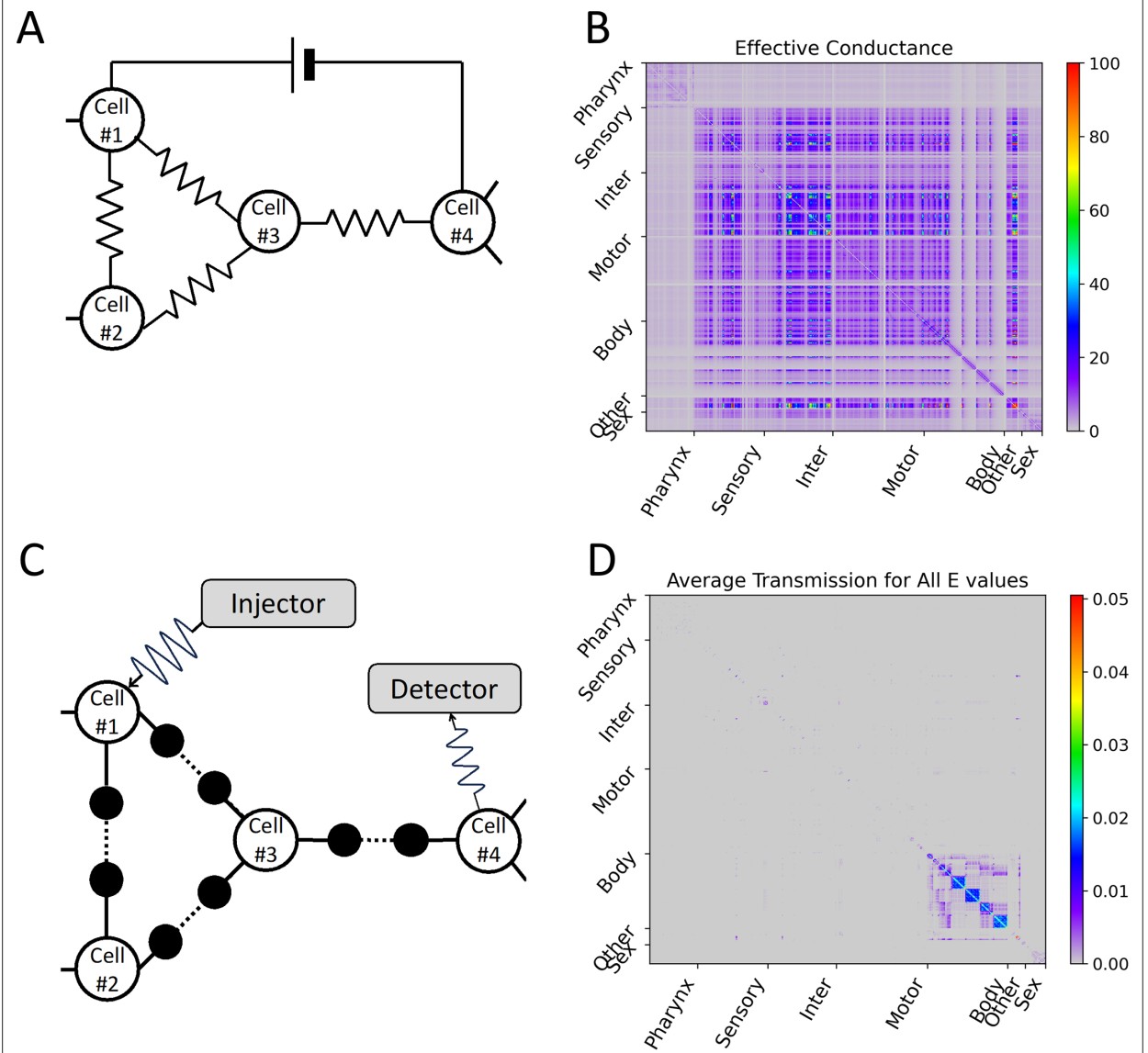

**Figure 5.** Differences depending on whether the electrical synapse network model considers interference. (**A**) In the electrical synapse network model without considering interference, when the electrical synapse is treated as an electrical resistor, the effective conductance (the reciprocal of the effective resistance) experienced by the external direct current power applied to an arbitrary cell-pair was calculated, and this value was represented as (**B**) a heatmap for all cell-pairs. (**C**) In our circuit considering interference of wave signals, the average transmission coefficient for all $E$ values (or all wavenumbers) of an arbitrary cell-pair was calculated, and this value was represented as (**D**) a heatmap for all cell-pairs. (**B, D**) In heatmap, the $X$ and $Y$ axes are arranged according to the index of 469 cells, and the index follows that of the original connectome data. Instead of displaying all cell names, only seven cell types are indicated.

anatomical gap junction network (***Cook et al., 2019***). The 469 cell names and the gap junction weight ($g_{ij}$) of all cell-pairs connected by gap junction were provided in the tab, and the weights had a value distribution from a minimum of 1 to a maximum of 401 (***Figure 1A***). The weights were proportional to the spatial size of the corresponding gap junction, which we hypothesized that the larger the weight, the stronger the connection between the cells and thus the better transmit the wave signal to the connected cell.

Our circuit model described *C. elegans'* anatomical gap junction network as follows (***Figure 1B***). First, each cell was treated as a node in our circuit. Each gap junction connected to a cell-pair was treated as an edge connecting two nodes in our circuit, and we inserted two virtual nodes (VNs) between those two nodes to be connected sequentially through VNs (Cell-VN-VN-Cell). Here, all edges inside our circuit were set to have the identical length $a$. The edge of Cell-to-VN (or VN-to-Cell)

representing like axon or dendrite of neuron gave a weight of 1, and the edge of VN-to-VN representing a gap junction gave our processed weight ($w_{ij} = \min\{1, g_{ij}/40\}$). Our processed weight ($w_{ij}$) was proportional to the anatomical gap junction weight ($g_{ij}$) and normalized as a value between 0 and 1 but was fixed to 1 if $g_{ij} > 40$. In our circuit model, the weight of the edge meant the transmission coefficient of the signal transmitted through this edge, so if the weight was 1 corresponded to complete transmission, and 0 corresponded to complete reflection (or unavailable edge), respectively. And within our circuit, the characteristic of the signal being spontaneously attenuated, and disappearing was excluded. Therefore, when a signal incoming from the outside to a node in our circuit was reflected to the outside at the same node and simultaneously transmitted to the outside at another node, the sum of the transmission and reflection coefficients to the outside always remained at 1 without loss.

## Calculation for the transmission coefficient

We benefited from the method of calculating the transmission coefficient according to the energy $E$ of the incoming wave when the electron probability wave incident on one point of the network lattice was detected on another point. This methodology, called the quantum percolation model, considered the propagation and interference of waves on the network lattice using the tight-binding Anderson Hamiltonian (*Anderson, 1958*; *Chang et al., 1995*; *Meir et al., 1989*; *Shapir et al., 1982*; *Thomas and Nakanishi, 2016*). In our methodology, we replaced the classical wave signals of complex number expression representing subthreshold membrane potential waves instead of the electron probability wave and replaced the network lattice with our circuit describing the electrical synapse network model built above. Then, the transmission coefficient calculated by the quantum percolation model told how well the subthreshold waves were transmitted from one cell $i$ to another cell $j$ while undergoing interference by multiple propagation paths in the electrical synapse network model. Here, the electrical synapses act as wave scatters that interrupt the propagation of the subthreshold waves from cell $i$ to cell $j$.

In the quantum percolation model, the tight-binding Anderson Hamiltonian was defined as follows:

$$H = \sum_m \epsilon_m |m\rangle \langle m| + \sum_{\langle mn \rangle} V_{mn} \left( |m\rangle \langle n| + |n\rangle \langle m| \right),$$

where the $|m\rangle$ (also expressed as the $\psi_m$) is as the electron probability wave on the $m$th node, and the $\langle m|$ means its complex conjugate $\psi_m^*$. We used $\psi_m$ as the classical wave signal observed on the $m$th node, which of complex number expression made it easy to express the phase shift of the wave and calculate the interference. The amplitude of this classical wave signal on the $m$th node was defined as the $\langle m|m\rangle = \psi_m^* \psi_m$. In the Hamiltonian, the $\epsilon_m$ is the on-site energy, and we regarded it as the resting membrane potential of the $m$th cell and set it to 0, the identical value as the relative reference value of the resting membrane potential in all cells. The $\langle mn \rangle$ represents a pair of nearest-neighbor sites in the Hamiltonian, we replaced it with a node-pair connected by the edges in our circuit. The $V_{mn}$ as a hopping matrix represents the entire structure of the network lattice in the Hamiltonian, which we implemented the structure of our circuit by setting it to $V_{mn} = 1$ for a node-pair between cell and VN, and $V_{mn} = w_{mn}$ for that between two VNs. Since the total number of the cells was 469, and we inserted two VNs for every 1433 electrical synapses, the newly constructed tight-binding Anderson Hamiltonian for our circuit was represented as a matrix of 3335 by 3335.

By solving the time-independent Schrödinger equation ($\vec{H\psi} = E\vec{\psi}$) for the Hamiltonian of our circuit, we sought to obtain the reflection and transmission coefficients of the cell-pair between the IN-cell and the OUT-cell from the $\psi_{IN}$ of the IN-cell where the wave signal from the outside was incident and reflect, and the $\psi_{OUT}$ of the OUT-cell where the wave signal from our circuit transmitted to the outside. We defined $\psi_{IN}$ and $\psi_{OUT}$ as follows using complex numbers $r$ and $t$ with amplitudes and phases,

$$\psi_{IN} = 1 + r, \psi_{OUT} = t,$$

where the $\psi_{IN}$ was the sum of 1 and $r$ corresponding to the incoming wave and the reflective wave, respectively, and the $\psi_{OUT}$ was $t$ corresponding to the transmitted wave. The reason why the amplitude

and the phase of the incoming wave on the IN-cell were 1 and 0, respectively, was because we set that as a reference amplitude and a reference phase in our circuit. The amplitude of the wave reflected from the IN-cell to the outside became the reflection coefficient with $R = r^*r$, and the amplitude of the wave transmitted from the OUT-cell to the outside became the transmission coefficient with $T = t^*t$, and $T + R = 1$ had to be always satisfied, as mentioned above.

This methodology also assumed both an external injector and an external detector. The external injector injected the generated wave into the network lattice and received the reflective wave from the network lattice. The external detector received the transmitted wave from the network lattice. The waves on the external injector $\phi_{Injector}$ and the external detector $\phi_{Detector}$ were extended from the definitions of $\psi_{IN}$ and $\psi_{OUT}$ above, respectively, and determined as follows:

$$\phi_{Injector} = 1e^{-ika} + re^{ika}, \; \phi_{Detector} = te^{ika}.$$

The external injector and detector were set to be connected to the IN-cell and OUT-cell by a perfect conducting wire with a length of $a$, respectively. The wave outgoing to the external injector (detector) from the IN-cell (OUT-cell) was determined by multiplying the phase difference of $e^{ika}$ to the wave on the IN-cell (OUT-cell), whereas the wave incoming to the IN-cell from the external injector was determined by multiplying the phase difference of $e^{-ika}$ to the wave on the IN-cell. Insert these terms into the time-independent Schrödinger equation for the Hamiltonian of our circuit, the following equation was derived describing the entire system consisting of our circuit and the external injector and detector,

$$\boldsymbol{H}\psi_m + \phi_{Injector}\delta_{m,IN} + \phi_{Detector}\delta_{m,OUT} = E\psi_m,$$

where the $E$ value on the right side representing the energy of the entire system was equivalent to the energy $E$ of the incoming wave from the external injector.

In the quantum percolation model, the energy $E$ of the incoming wave was defined as a tight-binding energy function as follows:

$$E = 2\gamma \cos(ka),$$

where $\gamma$ meant the binding energy between the nearest-neighbor atoms in the tight-binding model, but we had to set it to an arbitrary value. In this study, it was set to $\gamma = 5$, and the decision process was described in the below section. Applying this energy function, the above equation for the entire system was expressed as the matrices as follows:

$$\begin{pmatrix} e^{ika} - 2\gamma \cos(ka) & V_{IN,m} & V_{IN,OUT} \\ V_{n,IN} & V_{nm} - 2\gamma \cos(ka)\,\delta_{nm} & V_{n,OUT} \\ V_{OUT,IN} & V_{OUT,m} & e^{ika} - 2\gamma \cos(ka) \end{pmatrix} \begin{pmatrix} 1 + r \\ \psi_m \\ t \end{pmatrix} = \begin{pmatrix} e^{ika} - e^{-ika} \\ 0 \\ 0 \end{pmatrix},$$

where $m$ and $n$ meant the 3333 remaining nodes except two nodes for IN-cell and OUT-cell. For a wavenumber $k$ in the range of $[0, \pi/a]$ (or an $E$ value in the range of $[-2\gamma, 2\gamma]$), all components of both the left-hand square-matrix and the right-hand column-vector were fully expressed as complex numbers, and then the $r$ and $t$ values in the left-hand column-vector were obtained by matrix multiplication of the inverse matrix of the left-hand square-matrix and the right-hand column-vector. From squared absolute the $r$ and $t$ values, the reflection and transmission coefficients of the cell-pair between IN-cell and OUT-cell at the $E$ value (or the corresponding wavenumber $k$) were deduced, respectively.

## Searching for optimal gamma

To determine the appropriate $\gamma$ of the above energy function, we preliminarily calculated the transmission coefficient for all cell-pairs between the 469 cells under all four conditions of $\gamma$, and compared the average transmission coefficient graph as a function of the $E$ value $\left\langle T_{ij}(E) \right\rangle_{All\,ij\,cell-pairs}$ (**Appendix 1— figure 1**). Where four values of 1, 2.5, 5, and 10 were attempted for $\gamma$, and equally spaced a hundred values were used for the wavenumber $k$ in the range of $[0, \pi/a]$. According to the above energy function, the minimum/maximum range of the $E$ value differed as $[-2\gamma, 2\gamma]$, and as $\gamma$ was larger, the points of the $\left\langle T(E) \right\rangle$ graph were placed sparsely on the $X$-axis of the $E$ value. In each of the $\left\langle T(E) \right\rangle$ graphs

for these four conditions of $\gamma$, similarly high $\langle T(E) \rangle$ values occurred in the vicinity of $E = 0, \pm 1.4$. Based on this preliminary estimation, we confirmed that the occurrence of high $\langle T(E) \rangle$ values appeared as a function of the $E$ value regardless of $\gamma$. Therefore, we considered $\gamma$ as just determining the minimum/maximum range of the $E$ value.

The appropriate $\gamma$ we wanted in this study was large enough to provide a wide $E$ value range to check the signal mobility edge in the $\langle T(E) \rangle$ graph, and the appropriate $\gamma$ had to be small so that the points in the $\langle T(E) \rangle$ graph did not become too sparse. It was the condition of $\gamma = 5$ that satisfied this demand out of the four conditions of $\gamma$. Because the condition of $\gamma = 2.5$ was not sufficient to confirm the signal mobility edge, and the condition of $\gamma = 10$ had too sparse points along the $X$-axis.

## Error report on our calculation for the transmission coefficient

In principle, the sum of transmission coefficient $T_{ij}(E)$ and reflection coefficient $R_{ij}(E)$ of any cell-pair at any $E$ value always had to satisfy 1 because our circuit model did not consider spontaneous attenuation of the wave signal. However, in the calculating program we wrote and used for this study, the error occurred with the sum of the two coefficients exceeding 1 (by the tolerance was $10^{-5}$) for a total of 5213 cell-pairs in three specific $E$ values (45 cell-pairs at $E = -1.000$, 3836 cell-pairs at $E = -0.375$, and 1332 cell-pairs at $E = 1.000$). We performed the calculations on a total of 109,746 cell-pairs for a total of 801 $E$ values, so these errors correspond to 0.006% of the total calculation results. To eliminate the effect of these errors in this study, we post-corrected the two coefficients of the 5213 cell-pairs in the three specific $E$ values in which the error occurred to $T_{ij}(E) = 0, R_{ij}(E) = 1$.

## Calculation for effective conductance

An electrical resistor network model was constructed that mimics *C. elegans*' electrical synapse network, with the 469 cells replaced with electrical conducting nodes and the 1433 electrical synapses replaced with electrical resistors connecting these nodes. The effective resistance and its reciprocal, the effective conductance, when a direct current power source outside the electrical resistor network model contacted a cell by its positive pole, and another cell by its negative pole, respectively, were calculated (**Klein and Randić, 1993**). The effective conductance shows how well electric current without interference flows between the two cells contacted to the external power source.

We expressed the electric voltage at node $i$ as $\nu_i$, the electric current flowing from node $i$ to node $j$ as $\mu_{ij}$, the resistance of the electrical resistor between node $i$ and node $j$ as $\chi_{ij}$, the conductance of that as $\omega_{ij} \left( = \chi_{ij}^{-1} \right)$. The anatomical gap junction weights ($g_{ij}$) were used as the conductance value of the electrical resistors in the network, $\omega_{ij} = g_{ij}$, with arbitrary units.

By Ohm's law, each electrical resistor in the network satisfies the following equation,

$$\mu_{ij} = \chi_{ij}^{-1} \left( \nu_i - \nu_j \right) = \omega_{ij} \left( \nu_i - \nu_j \right).$$

The sum of the electric currents entering each node by Kirchhoff's current law satisfied the equation below,

$$\sum_{j \neq i} \mu_{ij} = \begin{cases} I, & at\ i = (+) \\ -I, & at\ i = (-) \\ 0, & \text{otherwise} \end{cases},$$

where $(+)$ and $(-)$ meant a node in which the positive and negative poles of the external power source were in contact, respectively, and the capital letter $I$ was the total electric current supplied by the external power source. When Ohm's law above was substituted on the left-hand of this equation, it was expressed as $\nu_i \sum_{j \neq i} \omega_{ij} - \sum_{j \neq i} \omega_{ij} \nu_j$, and then the equation was able to express in matrix and vector as follows:

$$\boldsymbol{L}^\omega \vec{\nu} = I \left( \vec{e}_{(+)} - \vec{e}_{(-)} \right),$$

where $L^\omega$ was the Laplacian matrix of the electrical resistors' conductance matrix ($\omega$), defined as $L_{ij}^\omega = \delta_{ij} \sum_l \omega_{il} - \omega_{ij}$, $\vec{\nu}$ was a column-vector with an electric voltage at all nodes as a component, and $\vec{e}_{(+)}$ ($\vec{e}_{(-)}$) was a unit column-vector of the identical dimension as $\vec{\nu}$, with only the component of the (+) node ((−) node) having a value of 1 and all other components having a value of 0. By applying the pseudo-inverse matrix of the Laplacian matrix ($L^{\omega+}$) on both sides of the equation, we obtained a particular solution as $\vec{\nu} = IL^{\omega+}\left(\vec{e}_{(+)} - \vec{e}_{(-)}\right)$.

The effective resistance applied between the positive and negative poles of the external power source was defined by Ohm's law of the electric voltage difference between the (+) and (−) nodes and the total electric current, as follows:

$$R^{\text{eff}} = I^{-1}\left(\nu_{(+)} - \nu_{(-)}\right) = I^{-1}\left(\vec{e}_{(+)} - \vec{e}_{(-)}\right)^T \vec{\nu}.$$

Substituting here the particular solution of $\vec{\nu}$ obtained above was as follows:

$$R^{\text{eff}} = \left(\vec{e}_{(+)} - \vec{e}_{(-)}\right)^T L^{\omega+}\left(\vec{e}_{(+)} - \vec{e}_{(-)}\right).$$

As a result, we were able to obtain the effective resistance for any one cell-pair by calculating the pseudo-inverse matrix of the Laplacian matrix ($L^{\omega+}$) from the electrical resistors' conductance matrix ($\omega$). The effective conductance was defined as the reciprocal of the effective resistance, $G^{\text{eff}} = \left(R^{\text{eff}}\right)^{-1}$.

## Acknowledgements

We appreciate two Nobel Laureates, Prof. Erwin Neher and Prof. Kurt Wuthrich, for their insightful discussions on the impact of this work. We have benefited a lot from their suggestions in the course of this study. Funding: This work was supported in part by both pCoE program of DGIST [grant number: 23-CoE-BT-01] and a grant from iPT, Korea.

## Additional information

### Funding

| Funder | Grant reference number | Author |
| --- | --- | --- |
| pCoE program of DGIST | 23-CoE-BT-01 | Iksoo Chang<br>Taegon Chung<br>Sangyeol Kim |

The funders had no role in study design, data collection, and interpretation, or the decision to submit the work for publication.

### Author contributions

Iksoo Chang, Conceptualization, Formal analysis, Supervision, Funding acquisition, Methodology, Writing – original draft, Writing – review and editing; Taegon Chung, Formal analysis, Visualization, Methodology; Sangyeol Kim, Conceptualization, Formal analysis, Supervision, Visualization, Methodology, Writing – original draft, Writing – review and editing

### Author ORCIDs

Sangyeol Kim ⓘ https://orcid.org/0009-0001-3200-9726

Reviewer #1 (Public review): https://doi.org/10.7554/eLife.99904.3.sa1
Reviewer #2 (Public review): https://doi.org/10.7554/eLife.99904.3.sa2
Author response https://doi.org/10.7554/eLife.99904.3.sa3

# Additional files

**Supplementary files**
MDAR checklist

**Data availability**
All data generated or analyzed during this study are included in the manuscript and supporting files.

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

# Appendix 1

## Auxiliary paragraph 1

The 31 specific *E* values in this study were not fixed and were of a property that can change as a researcher adjusts the resolution of the *E* value. Perhaps the higher the resolution of the *E* value, the more specific *E* values will be found. And it was arbitrary that our reference value for strong transmission was 0.5, and this reference value can be raised or lowered depending on the researcher's intention, and the lower the reference value, the greater the number of cell-pairs with strong transmission will be found.

## Auxiliary paragraph 2

Because of the omitted the rest of cells and their electrical synapses in the network diagram, some nodes in the diagram look like remote islands, but it should be remembered that the 173 cells were connected as one large cluster through both the drawn and the omitted electrical synapses.

## Auxiliary paragraph 3

Because the number of the cell-pairs of intra-body-wall muscles was overwhelmingly large and it was relatively easy to measure the distance compared to those of other cell-pairs, the investigation for the distance excluded them (e.g., the distance of two body-wall muscles in the same strand easily calculate from the given number in the name of the muscles).

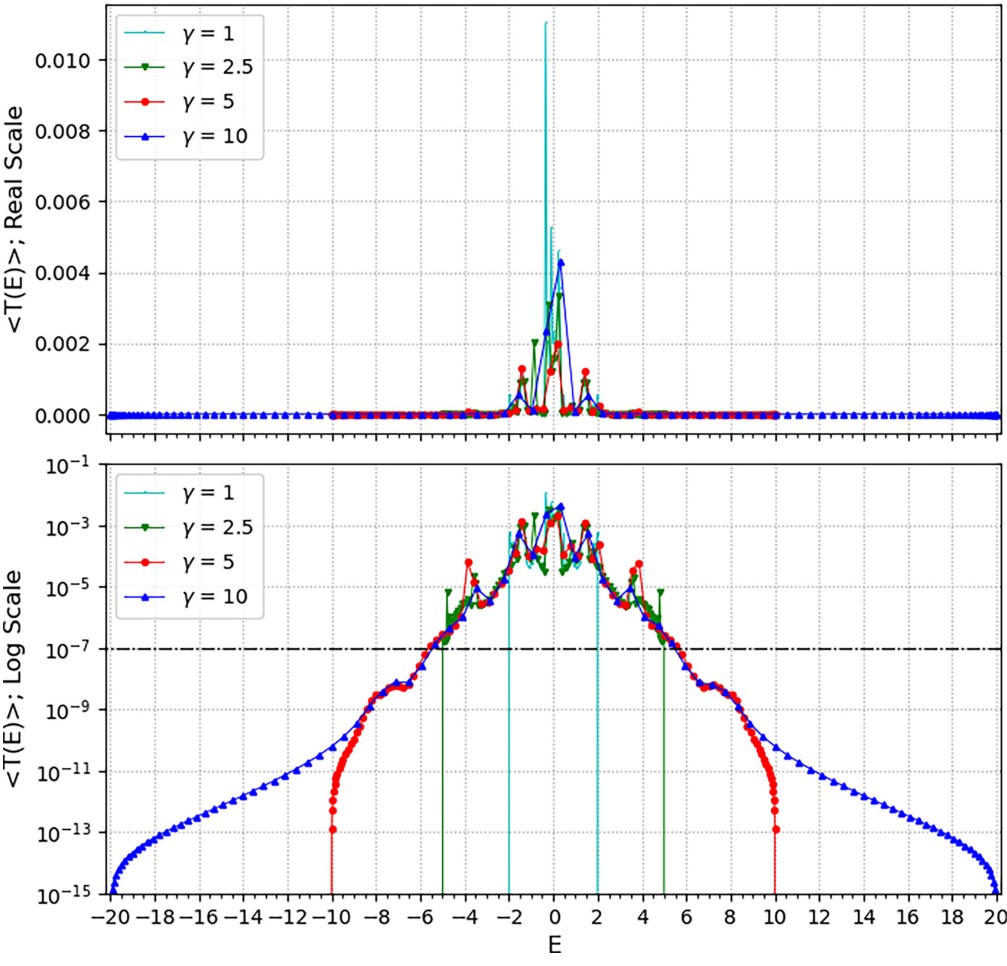

**Appendix 1—figure 1.** Searching for optimal gamma. The average transmission coefficient for all cell-pairs as a function of *E* value (*E* value is a function of both $\gamma$ and wavenumber *k*) was calculated under the four $\gamma$ conditions (1, 2.5, 5, and 10), respectively. Here, the wavenumbers used the same set as a hundred values equally spaced in the range of [0, $\pi/a$]. The upper panel shows the average transmission coefficient graphs as the *Y*-axis of the real scale and the lower panel shows the same graphs as the *Y*-axis of the log scale. The reference value ($10^{-7}$) of the signal mobility edge is shown in the lower panel as a dot-dash line.

**Appendix 1—table 1.** Cell names corresponding to the indices in the heatmaps for wavenumber-dependent transmission map.

Each two columns shows the index and the corresponding cell name of the cells that belonged to the cell-pairs with strong transmission at the $E$ value (or the corresponding wavenumber) indicated in the first row. The color painted on the name box stands for the cell type (red: pharynx cells, orange: sensory neurons, yellow: inter neurons, green: motor neurons, light blue: body-wall muscles, purple: other end organs, magenta: sex-specific cells).

| E = 0.000, ka = 1.571 | | E = 0.100, ka = 1.561 | | E = 0.175, ka = 1.553 | | E = 0.225, ka = 1.548 | | E = 0.275, ka = 1.543 | | E = 0.325, ka = 1.538 | | E = 0.475, ka = 1.523 | | E = 0.575, ka = 1.513 | |
| Index | Cell name | Index | Cell name | Index | Cell name | Index | Cell name | Index | Cell name | Index | Cell name | Index | Cell name | Index | Cell name |
|---|---|---|---|---|---|---|---|---|---|---|---|---|---|---|---|
| 0 | AVKR | 0 | dBWML7 | 0 | dBWML4 | 0 | IL1DR | 0 | vBWML4 | 0 | vBWML1 | 0 | PVPL | 0 | RIVL |
| 1 | SMBDR | 1 | dBWMR2 | 1 | dBWML5 | 1 | IL1L | 1 | vBWMR3 | 1 | vBWML3 | 1 | DVC | 1 | RIVR |
| 2 | dBWML4 | 2 | dBWMR3 | 2 | vBWMR1 | 2 | IL1R | 2 | vBWMR4 | 2 | vBWML4 | 2 | vBWML7 | | |
| 3 | dBWML5 | 3 | vBWML2 | 3 | vBWMR2 | 3 | IL1VL | 3 | vBWMR5 | 3 | vBWML5 | 3 | vBWML8 | | |
| 4 | dBWML8 | 4 | vBWML6 | 4 | vBWMR3 | 4 | IL1VR | 4 | vBWML8 | 4 | vBWML6 | 4 | hmc | | |
| 5 | dBWML10 | 5 | vBWMR1 | 5 | vBWMR4 | 5 | URAVR | 5 | vBWML10 | 5 | vBWML7 | | | | |
| 6 | dBWML11 | 6 | vBWMR2 | 6 | vBWMR5 | 6 | dBWMR5 | 6 | vBWML11 | 6 | vBWMR3 | | | | |
| 7 | dBWML12 | 7 | vBWMR3 | 7 | vBWMR6 | 7 | dBWMR6 | 7 | vBWML12 | 7 | vBWMR4 | | | | |
| 8 | dBWML13 | 8 | vBWMR4 | 8 | vBWMR7 | 8 | dBWMR7 | 8 | vBWML13 | 8 | vBWMR5 | | | | |
| 9 | dBWML14 | 9 | vBWMR5 | 9 | dBWMR8 | 9 | vBWML2 | 9 | vBWML15 | 9 | vBWMR6 | | | | |
| 10 | dBWML15 | 10 | vBWMR6 | 10 | dBWMR9 | 10 | vBWML3 | 10 | vBWML16 | 10 | vBWMR7 | | | | |
| 11 | dBWML17 | 11 | vBWMR7 | 11 | dBWMR10 | 11 | vBWML4 | 11 | vBWML17 | 11 | vBWML10 | | | | |
| 12 | dBWML18 | 12 | dBWML8 | 12 | dBWMR11 | 12 | vBWML6 | 12 | vBWML22 | 12 | vBWML11 | | | | |
| 13 | dBWML20 | 13 | vBWML9 | 13 | dBWMR12 | 13 | vBWML7 | 13 | vBWML23 | 13 | vBWML12 | | | | |
| 14 | dBWML21 | 14 | vBWML10 | 14 | dBWMR13 | 14 | vBWMR3 | 14 | vBWMR10 | 14 | vBWML16 | | | | |
| 15 | dBWML23 | 15 | vBWML11 | 15 | dBWMR14 | 15 | vBWMR4 | 15 | vBWMR11 | 15 | vBWML17 | | | | |
| 16 | dBWML24 | 16 | vBWML12 | 16 | dBWMR15 | 16 | vBWMR6 | 16 | vBWMR12 | 16 | vBWML18 | | | | |
| 17 | vBWML11 | 17 | vBWML13 | 17 | dBWMR16 | 17 | vBWMR7 | 17 | vBWMR14 | 17 | vBWML19 | | | | |
| 18 | vBWML14 | 18 | vBWML15 | 18 | dBWMR17 | 18 | dBWMR9 | 18 | vBWMR15 | 18 | vBWML21 | | | | |
| 19 | vBWML17 | 19 | vBWML16 | 19 | dBWMR18 | 19 | dBWMR10 | 19 | vBWMR16 | 19 | vBWML22 | | | | |
| 20 | CANL | 20 | vBWML17 | 20 | dBWMR19 | 20 | dBWMR11 | 20 | vBWMR19 | 20 | vBWML23 | | | | |

*Appendix 1—table 1 continued on next page*

Appendix 1—table 1 continued

| E = 0.000 | ka = 1.571 | E = 0.100 | ka = 1.561 | E = 0.175 | ka = 1.553 | E = 0.225 | ka = 1.548 | E = 0.275 | ka = 1.543 | E = 0.325 | ka = 1.538 | E = 0.475 | ka = 1.523 | E = 0.575 | ka = 1.513 |
|---|---|---|---|---|---|---|---|---|---|---|---|---|---|---|---|
| 21 | CANR | 21 | vBWML18 | 21 | dBWMR20 | 21 | dBWMR13 | 21 | vBWMR20 | 21 | vBWMR9 | | | | |
| 22 | exc_cell | 22 | vBWML19 | 22 | dBWMR21 | 22 | dBWMR14 | 22 | vBWMR21 | 22 | vBWMR10 | | | | |
| 23 | hmc | 23 | vBWML20 | 23 | dBWMR23 | 23 | dBWMR16 | 23 | vBWMR23 | 23 | vBWMR11 | | | | |
| | | 24 | vBWML21 | 24 | dBWMR24 | 24 | dBWMR17 | 24 | vBWMR24 | 24 | vBWMR12 | | | | |
| | | 25 | vBWML22 | 25 | vBWML9 | 25 | dBWMR18 | 25 | CANL | 25 | vBWMR13 | | | | |
| | | 26 | vBWML23 | 26 | vBWML11 | 26 | dBWMR20 | 26 | CANR | 26 | vBWMR16 | | | | |
| | | 27 | vBWMR8 | 27 | vBWML12 | 27 | dBWMR21 | 27 | exc_cell | 27 | vBWMR17 | | | | |
| | | 28 | vBWMR9 | 28 | vBWML14 | 28 | dBWMR23 | 28 | hmc | 28 | vBWMR18 | | | | |
| | | 29 | vBWMR10 | 29 | vBWML15 | 29 | dBWMR24 | | | 29 | vBWMR19 | | | | |
| | | 30 | vBWMR11 | 30 | vBWML17 | 30 | vBWML8 | | | 30 | vBWMR20 | | | | |
| | | 31 | vBWMR12 | 31 | vBWML18 | 31 | vBWML9 | | | 31 | vBWMR22 | | | | |
| | | 32 | vBWMR13 | 32 | vBWMR9 | 32 | vBWML10 | | | 32 | vBWMR23 | | | | |
| | | 33 | vBWMR14 | 33 | vBWMR12 | 33 | vBWML11 | | | 33 | vBWMR24 | | | | |
| | | 34 | vBWMR15 | 34 | vBWMR15 | 34 | vBWML13 | | | 34 | CANL | | | | |
| | | 35 | vBWMR16 | 35 | vBWMR18 | 35 | vBWML14 | | | 35 | CANR | | | | |
| | | 36 | vBWMR17 | 36 | vBWMR21 | 36 | vBWML16 | | | 36 | exc_cell | | | | |
| | | 37 | vBWMR18 | 37 | vBWMR24 | 37 | vBWML17 | | | 37 | hmc | | | | |
| | | 38 | vBWMR19 | 38 | CANL | 38 | vBWML18 | | | | | | | | |
| | | 39 | vBWMR20 | 39 | CANR | 39 | vBWMR9 | | | | | | | | |
| | | 40 | vBWMR21 | 40 | exc_cell | 40 | vBWMR10 | | | | | | | | |
| | | 41 | vBWMR22 | | | 41 | vBWMR11 | | | | | | | | |
| | | 42 | vBWMR23 | | | 42 | vBWMR13 | | | | | | | | |
| | | 43 | vBWMR24 | | | 43 | vBWMR14 | | | | | | | | |
| | | 44 | CANL | | | 44 | vBWMR16 | | | | | | | | |
| | | 45 | CANR | | | 45 | vBWMR17 | | | | | | | | |

Appendix 1—table 1 continued

| E = 0.000 | ka = 1.571 | E = 0.100 | ka = 1.561 | E = 0.175 | ka = 1.553 | E = 0.225 | ka = 1.548 | E = 0.275 | ka = 1.543 | E = 0.325 | ka = 1.538 | E = 0.475 | ka = 1.523 | E = 0.575 | ka = 1.513 |
|---|---|---|---|---|---|---|---|---|---|---|---|---|---|---|---|
| | | | | | | 46 | vBWMRl8 | | | | | | | | |
| | | | | | | 47 | vBWMR20 | | | | | | | | |
| | | | | | | 48 | vBWMR21 | | | | | | | | |
| | | | | | | 49 | vBWMR23 | | | | | | | | |
| | | | | | | 50 | vBWMR24 | | | | | | | | |
| | | | | | | 51 | CANL | | | | | | | | |
| | | | | | | 52 | CANR | | | | | | | | |
| | | | | | | 53 | exc_cell | | | | | | | | |
| | | | | | | 54 | hmc | | | | | | | | |

**Appendix 1—table 2.** Cell names corresponding to the indices in the heatmaps for wavenumber-dependent transmission map. It is represented in the same way as **Appendix 1—table 1.**

| E = 0.750 | ka = 1.496 | E = 0.800 | ka = 1.491 | E = 0.850 | ka = 1.486 | E = 0.925 | ka = 1.478 | E = 1.350 | ka = 1.435 | E = 1.450 | ka = 1.425 | E = 1.500 | ka = 1.420 | E = 1.575 | ka = 1.413 |
|---|---|---|---|---|---|---|---|---|---|---|---|---|---|---|---|
| Index | Cell name | Index | Cell name | Index | Cell name | Index | Cell name | Index | Cell name | Index | Cell name | Index | Cell name | Index | Cell name |
| 0 | PVDL | 0 | PVDL | 0 | PVDL | 0 | RIGL | 0 | dBWML3 | 0 | dBWML3 | 0 | dBWML9 | 0 | dBWML10 |
| 1 | PVDR | 1 | PVDR | 1 | PVDR | 1 | RIGR | 1 | dBWML6 | 1 | dBWML4 | 1 | dBWML10 | 1 | dBWML11 |
| 2 | ALA | 2 | ALA | 2 | ALA | | | 2 | dBWML9 | 2 | dBWML5 | 2 | dBWML11 | 2 | dBWML12 |
| 3 | dBWML7 | 3 | dBWML7 | 3 | dBWML7 | | | 3 | dBWML10 | 3 | dBWML6 | 3 | dBWML12 | 3 | dBWML13 |
| 4 | dBWML8 | 4 | dBWMR7 | 4 | dBWML8 | | | 4 | dBWML11 | 4 | dBWMR1 | 4 | dBWML13 | 4 | dBWML14 |
| 5 | CANL | 5 | vBWML3 | | | | | 5 | dBWML12 | 5 | dBWMR3 | 5 | dBWML14 | 5 | dBWML15 |
| 6 | CANR | 6 | vBWML5 | | | | | 6 | dBWML13 | 6 | vBWML1 | 6 | dBWML15 | 6 | dBWML16 |
| | | 7 | vBWML7 | | | | | 7 | dBWML14 | 7 | vBWML3 | 7 | dBWML16 | 7 | dBWML17 |
| | | 8 | vBWMR6 | | | | | 8 | dBWML15 | 8 | vBWML5 | 8 | dBWML17 | 8 | dBWML18 |
| | | 9 | vBWMR7 | | | | | 9 | dBWML16 | 9 | vBWML7 | 9 | dBWML18 | 9 | dBWML19 |
| | | 10 | dBWML8 | | | | | 10 | dBWML17 | 10 | vBWMR1 | 10 | dBWML19 | 10 | dBWML20 |
| | | 11 | dBWMR8 | | | | | 11 | dBWML18 | 11 | vBWMR2 | 11 | dBWML20 | 11 | dBWML21 |
| | | 12 | dBWMR9 | | | | | 12 | dBWML19 | 12 | vBWMR3 | 12 | dBWML21 | 12 | dBWML22 |
| | | 13 | vBWML8 | | | | | 13 | dBWML20 | 13 | vBWMR5 | 13 | dBWML22 | 13 | dBWML23 |
| | | 14 | vBWMR8 | | | | | 14 | dBWML21 | 14 | vBWMR7 | 14 | dBWML23 | 14 | dBWML24 |
| | | 15 | exc_gl | | | | | 15 | dBWML22 | 15 | dBWML9 | 15 | dBWML24 | 15 | dBWMR9 |
| | | | | | | | | 16 | dBWML23 | 16 | dBWML10 | 16 | dBWMR8 | 16 | dBWMR10 |
| | | | | | | | | 17 | dBWML24 | 17 | dBWML11 | 17 | dBWMR9 | 17 | dBWMR11 |
| | | | | | | | | 18 | dBWMR8 | 18 | dBWML12 | 18 | dBWMR10 | 18 | dBWMR12 |
| | | | | | | | | 19 | dBWMR9 | 19 | dBWML13 | 19 | dBWMR11 | 19 | dBWMR13 |
| | | | | | | | | 20 | dBWMR10 | 20 | dBWML14 | 20 | dBWMR12 | 20 | dBWMR14 |
| | | | | | | | | 21 | dBWMR11 | 21 | dBWML15 | 21 | dBWMR13 | 21 | dBWMR15 |
| | | | | | | | | 22 | dBWMR12 | 22 | dBWML16 | 22 | dBWMR14 | 22 | dBWMR16 |

Appendix 1—table 2 continued

| E = 0.750 | ka = 1.496 | E = 0.800 | ka = 1.491 | E = 0.850 | ka = 1.486 | E = 0.925 | ka = 1.478 | E = 1.350 | ka = 1.435 | E = 1.450 | ka = 1.425 | E = 1.500 | ka = 1.420 | E = 1.575 | ka = 1.413 |
|---|---|---|---|---|---|---|---|---|---|---|---|---|---|---|---|
|  |  |  |  |  |  |  |  | 23 | dBWMR13 | 23 | dBWML17 | 23 | dBWMR15 | 23 | dBWMR17 |
|  |  |  |  |  |  |  |  | 24 | dBWMR14 | 24 | dBWML18 | 24 | dBWMR16 | 24 | dBWMR18 |
|  |  |  |  |  |  |  |  | 25 | dBWMR15 | 25 | dBWML19 | 25 | dBWMR17 | 25 | dBWMR19 |
|  |  |  |  |  |  |  |  | 26 | dBWMR16 | 26 | dBWML20 | 26 | dBWMR18 | 26 | dBWMR20 |
|  |  |  |  |  |  |  |  | 27 | dBWMR17 | 27 | dBWML21 | 27 | dBWMR19 | 27 | dBWMR21 |
|  |  |  |  |  |  |  |  | 28 | dBWMR18 | 28 | dBWML22 | 28 | dBWMR20 | 28 | dBWMR22 |
|  |  |  |  |  |  |  |  | 29 | dBWMR19 | 29 | dBWML23 | 29 | dBWMR21 | 29 | dBWMR23 |
|  |  |  |  |  |  |  |  | 30 | dBWMR20 | 30 | dBWML24 | 30 | dBWMR22 | 30 | dBWMR24 |
|  |  |  |  |  |  |  |  | 31 | dBWMR21 | 31 | dBWMR8 | 31 | dBWMR23 | 31 | vBWML9 |
|  |  |  |  |  |  |  |  | 32 | dBWMR22 | 32 | dBWMR9 | 32 | dBWMR24 | 32 | vBWML10 |
|  |  |  |  |  |  |  |  | 33 | dBWMR23 | 33 | dBWMR10 | 33 | vBWML9 | 33 | vBWML11 |
|  |  |  |  |  |  |  |  | 34 | dBWMR24 | 34 | dBWMR11 | 34 | vBWML10 | 34 | vBWML12 |
|  |  |  |  |  |  |  |  | 35 | vBWML9 | 35 | dBWMR13 | 35 | vBWML11 | 35 | vBWML13 |
|  |  |  |  |  |  |  |  | 36 | vBWML10 | 36 | dBWMR14 | 36 | vBWML12 | 36 | vBWML14 |
|  |  |  |  |  |  |  |  | 37 | vBWML11 | 37 | dBWMR15 | 37 | vBWML16 | 37 | vBWML15 |
|  |  |  |  |  |  |  |  | 38 | vBWML12 | 38 | dBWMR16 | 38 | vBWML17 | 38 | vBWML16 |
|  |  |  |  |  |  |  |  | 39 | vBWML13 | 39 | dBWMR17 | 39 | vBWML18 | 39 | vBWML17 |
|  |  |  |  |  |  |  |  | 40 | vBWML14 | 40 | dBWMR18 | 40 | vBWML19 | 40 | vBWML18 |
|  |  |  |  |  |  |  |  | 41 | vBWML15 | 41 | dBWMR19 | 41 | vBWMR9 | 41 | vBWML19 |
|  |  |  |  |  |  |  |  | 42 | vBWML16 | 42 | dBWMR20 | 42 | vBWMR10 | 42 | vBWML23 |
|  |  |  |  |  |  |  |  | 43 | vBWML17 | 43 | dBWMR21 | 43 | vBWMR11 | 43 | vBWMR9 |
|  |  |  |  |  |  |  |  | 44 | vBWML18 | 44 | dBWMR22 | 44 | vBWMR12 | 44 | vBWMR10 |
|  |  |  |  |  |  |  |  | 45 | vBWML20 | 45 | dBWMR23 | 45 | vBWMR13 | 45 | vBWMR11 |
|  |  |  |  |  |  |  |  | 46 | vBWML22 | 46 | dBWMR24 | 46 | vBWMR14 | 46 | vBWMR12 |
|  |  |  |  |  |  |  |  | 47 | vBWML23 | 47 | vBWML9 | 47 | vBWMR16 | 47 | vBWMR13 |
|  |  |  |  |  |  |  |  | 48 | vBWMR9 | 48 | vBWML10 | 48 | vBWMR17 | 48 | vBWMR14 |

*Appendix 1—table 2 continued*

| E = 0.750 | ka = 1.496 | E = 0.800 | ka = 1.491 | E = 0.850 | ka = 1.486 | E = 0.925 | ka = 1.478 | E = 1.350 | ka = 1.435 | E = 1.450 | ka = 1.425 | E = 1.500 | ka = 1.420 | E = 1.575 | ka = 1.413 |
|---|---|---|---|---|---|---|---|---|---|---|---|---|---|---|---|
| | | | | | | | | 49 | vBWMR10 | 49 | vBWML11 | 49 | vBWMR18 | 49 | vBWMR15 |
| | | | | | | | | 50 | vBWMR11 | 50 | vBWML12 | 50 | vBWMR19 | 50 | vBWMR16 |
| | | | | | | | | 51 | vBWMR12 | 51 | vBWML14 | 51 | vBWMR20 | 51 | vBWMR17 |
| | | | | | | | | 52 | vBWMR13 | 52 | vBWML16 | 52 | vBWMR21 | 52 | vBWMR18 |
| | | | | | | | | 53 | vBWMR14 | 53 | vBWML17 | 53 | vBWMR23 | 53 | vBWMR19 |
| | | | | | | | | 54 | vBWMR15 | 54 | vBWML18 | 54 | vBWMR24 | 54 | vBWMR20 |
| | | | | | | | | 55 | vBWMR16 | 55 | vBWML19 | | | 55 | vBWMR21 |
| | | | | | | | | 56 | vBWMR17 | 56 | vBWML20 | | | 56 | vBWMR22 |
| | | | | | | | | 57 | vBWMR18 | 57 | vBWML21 | | | 57 | vBWMR23 |
| | | | | | | | | 58 | vBWMR19 | 58 | vBWML22 | | | 58 | vBWMR24 |
| | | | | | | | | 59 | vBWMR20 | 59 | vBWML23 | | | | |
| | | | | | | | | 60 | vBWMR21 | 60 | vBWMR9 | | | | |
| | | | | | | | | 61 | vBWMR22 | 61 | vBWMR11 | | | | |
| | | | | | | | | 62 | vBWMR23 | 62 | vBWMR12 | | | | |
| | | | | | | | | 63 | vBWMR24 | 63 | vBWMR13 | | | | |
| | | | | | | | | | | 64 | vBWMR14 | | | | |
| | | | | | | | | | | 65 | vBWMR15 | | | | |
| | | | | | | | | | | 66 | vBWMR16 | | | | |
| | | | | | | | | | | 67 | vBWMR17 | | | | |
| | | | | | | | | | | 68 | vBWMR18 | | | | |
| | | | | | | | | | | 69 | vBWMR19 | | | | |
| | | | | | | | | | | 70 | vBWMR20 | | | | |
| | | | | | | | | | | 71 | vBWMR21 | | | | |
| | | | | | | | | | | 72 | vBWMR22 | | | | |
| | | | | | | | | | | 73 | vBWMR23 | | | | |
| | | | | | | | | | | 74 | vBWMR24 | | | | |

**Appendix 1—table 3.** Cell names corresponding to the indices in the heatmaps for wavenumber-dependent transmission map. It is represented in the same way as *Appendix 1—table 1*.

| E = 1.700 Index | ka = 1.400 Cell name | E = 1.750 Index | ka = 1.395 Cell name | E = 1.800 Index | ka = 1.390 Cell name | E = 1.875 Index | ka = 1.382 Cell name | E = 2.000 Index | ka = 1.369 Cell name | E = 2.225 Index | ka = 1.346 Cell name | E = 2.450 Index | ka = 1.323 Cell name | E = 2.650 Index | ka = 1.303 Cell name |
|---|---|---|---|---|---|---|---|---|---|---|---|---|---|---|---|
| 0 | vm2AL | 0 | dBWMR1 | 0 | vBWML4 | 0 | dBWMR7 | 0 | dBWMR1 | 0 | dBWMR4 | 0 | pm5VL | 0 | RMEL |
| 1 | vm2AR | 1 | dBWMR2 | 1 | vBWML6 | 1 | vBWML4 | 1 | dBWMR2 | 1 | dBWMR6 | 1 | mc1V | 1 | RMED |
| 2 | vm2PL | 2 | dBWMR4 | 2 | vBWMR1 | 2 | vBWML6 | 2 | dBWMR3 | | | 2 | ASEL | 2 | dBWMR1 |
| 3 | vm2PR | 3 | vBWML8 | 3 | vBWMR2 | 3 | vBWMR6 | 3 | dBWMR7 | | | 3 | AIZL | 3 | GLRL |
| | | 4 | vBWML10 | 4 | vBWMR6 | 4 | dBWMR9 | 4 | GLRDL | | | | | 4 | GLRR |
| | | 5 | vm1AL | 5 | dBWMR9 | 5 | vBWML8 | 5 | GLRDR | | | | | | |
| | | 6 | vm1AR | 6 | hmc | 6 | vBWMR8 | 6 | hmc | | | | | | |
| | | 7 | vm1PL | | | | | | | | | | | | |
| | | 8 | vm1PR | | | | | | | | | | | | |

**Appendix 1—table 4.** Cell names corresponding to the indices in the heatmaps for wavenumber-dependent transmission map. It is represented in the same way as **Appendix 1—table 1**.

| E = 3.000 ka = 1.266 | | E = 3.375 ka = 1.227 | | E = 3.475 ka = 1.216 | | E = 3.525 ka = 1.211 | | E = 3.625 ka = 1.200 | | E = 3.750 ka = 1.186 | | E = 3.875 ka = 1.173 | |
|---|---|---|---|---|---|---|---|---|---|---|---|---|---|
| Index | Cell name | Index | Cell name | Index | Cell name | Index | Cell name | Index | Cell name | Index | Cell name | Index | Cell name |
| 0 | PLNR | 0 | ALA | 0 | PVDR | 0 | PVDL | 0 | ASKR | 0 | DVB | 0 | PHCR |
| 1 | PHAL | 1 | hmc | 1 | hmc | 1 | PVDR | 1 | PVQL | 1 | DD06 | 1 | PVR |
| 2 | PHCL | | | | | | | | | | | 2 | DVB |
| 3 | DA09 | | | | | | | | | | | 3 | PVT |
| 4 | PDB | | | | | | | | | | | 4 | AVL |
| | | | | | | | | | | | | 5 | PDA |
| | | | | | | | | | | | | 6 | PDB |
| | | | | | | | | | | | | 7 | VA12 |
| | | | | | | | | | | | | 8 | VD12 |

**Appendix 1—table 5.** List of cell-pairs with strong transmission except intra body-wall muscles pairs. The first column represents the shortest distance along the network connecting the two cells of the cell-pair. The second and third columns represent the cell names of the cells of the cell-pair, and the color painted on the name box stands for the cell type (red: pharynx cells, orange: sensory neurons, yellow: inter neurons, green: motor neurons, light blue: body-wall muscles, purple: other end organs, magenta: sex-specific cells). The fourth, fifth, and sixth columns represent the transmission coefficient, the E value, and the wavenumber when the cell-pair shows the strongest transmission in the positive E value range, respectively.

| Distance | Cell names | Cell names | Max{T(E)} | E(ka) | ka | Distance | Cell names | Cell names | Max{T(E)} | E(ka) | ka | Cell names | Cell names | Max{T(E)} | E(ka) | ka |
|---|---|---|---|---|---|---|---|---|---|---|---|---|---|---|---|---|
| 17 | dBWMR24 | hmc | 0.912 | 0.125 | 1.558 | 3 | vBWMR10 | hmc | 0.791 | 0.325 | 1.538 | vm2AR | vm2PR | 0.892 | 1.700 | 1.400 |
| 17 | vBWMR24 | hmc | 0.765 | 0.325 | 1.538 | 3 | um1AL | vm2PR | 0.659 | 2.025 | 1.367 | ASEL | AFDL | 0.794 | 0.125 | 1.558 |
| 16 | vBWMR23 | hmc | 0.797 | 0.325 | 1.538 | 3 | um1AR | vm2PL | 0.659 | 2.025 | 1.367 | PVDL | PVDR | 0.858 | 0.750 | 1.496 |
| 15 | dBWMR22 | hmc | 0.805 | 0.125 | 1.558 | 3 | um1PL | vm2AR | 0.659 | 2.025 | 1.367 | PVDL | ALA | 0.704 | 0.850 | 1.486 |
| 15 | vBWMR22 | hmc | 0.603 | 0.325 | 1.538 | 3 | um1PR | vm2AL | 0.659 | 2.025 | 1.367 | PVDL | CANL | 0.776 | 0.750 | 1.496 |
| 14 | dBWMR21 | hmc | 0.946 | 0.125 | 1.558 | 2 | pm5VL | mc1V | 0.665 | 2.450 | 1.323 | PVDR | ALA | 0.728 | 0.750 | 1.496 |
| 12 | dBWMR19 | hmc | 0.892 | 0.125 | 1.558 | 2 | ASEL | AIZL | 0.839 | 2.450 | 1.323 | PVDR | CANR | 0.914 | 0.750 | 1.496 |
| 12 | vBWMR19 | hmc | 0.692 | 0.325 | 1.538 | 2 | ASKR | PVQL | 0.746 | 3.625 | 1.200 | PVDR | hmc | 0.627 | 3.475 | 1.216 |
| 11 | dBWMR18 | hmc | 0.731 | 0.125 | 1.558 | 2 | PLNR | PDB | 0.802 | 3.000 | 1.266 | PHBL | PHBR | 0.852 | 0.375 | 1.533 |
| 11 | vBWMR18 | hmc | 0.754 | 0.325 | 1.538 | 2 | PVDL | dBWML7 | 0.946 | 0.850 | 1.486 | CEPDL | OLQDL | 0.639 | 0.050 | 1.566 |
| 10 | dBWMR17 | hmc | 0.645 | 0.125 | 1.558 | 2 | PVDL | dBWML8 | 0.947 | 0.850 | 1.486 | IL1DR | IL1R | 0.642 | 0.225 | 1.548 |
| 10 | vBWMR17 | hmc | 0.789 | 0.325 | 1.538 | 2 | PVDL | CANR | 0.851 | 0.750 | 1.496 | IL1L | IL1VL | 0.599 | 0.225 | 1.548 |
| 9 | dBWMR16 | hmc | 0.936 | 0.125 | 1.558 | 2 | PVDR | dBWML7 | 0.737 | 0.850 | 1.486 | IL1R | IL1VR | 0.750 | 0.225 | 1.548 |
| 9 | vBWMR16 | hmc | 0.739 | 0.325 | 1.538 | 2 | PVDR | dBWML8 | 0.738 | 0.850 | 1.486 | IL1VL | IL1VL | 0.973 | 0.225 | 1.548 |
| 8 | vBWML15 | hmc | 0.593 | 0.275 | 1.543 | 2 | PVDR | CANL | 0.895 | 0.750 | 1.496 | IL1VR | URAVR | 0.556 | 0.225 | 1.548 |
| 7 | AVBR | dBWML12 | 0.601 | 1.000 | 1.471 | 2 | PHAL | DA09 | 0.546 | 3.000 | 1.266 | AIZL | AIZR | 0.534 | 0.375 | 1.533 |
| 7 | dBWML14 | hmc | 0.867 | 0.125 | 1.558 | 2 | PHAL | PDB | 0.662 | 3.000 | 1.266 | ALA | CANR | 0.594 | 0.750 | 1.496 |
| 7 | vBWML14 | hmc | 0.726 | 0.225 | 1.548 | 2 | PHCL | PDB | 0.577 | 3.000 | 1.266 | PVPL | DVC | 0.568 | 0.475 | 1.523 |
| 7 | vBWML16 | hmc | 0.776 | 0.275 | 1.543 | 2 | IL1DR | IL1L | 0.781 | 0.225 | 1.548 | DVB | DD06 | 0.535 | 3.750 | 1.186 |
| 6 | dBWMR13 | hmc | 0.900 | 0.125 | 1.558 | 2 | IL1DR | IL1VR | 0.844 | 1.425 | 1.428 | RIGL | RIGR | 0.760 | 0.925 | 1.478 |
| 6 | vBWML13 | hmc | 0.832 | 0.225 | 1.548 | 2 | IL1L | IL1VR | 0.511 | 0.225 | 1.548 | AVKR | SMBDR | 0.895 | 0.200 | 1.551 |

*Appendix 1—table 5 continued*

| Distance | Cell names | Max{T(E)} | E(ka) | ka | Distance | Cell names | Max{T(E)} | E(ka) | ka | Cell names | Max{T(E)} | E(ka) | ka |
|---|---|---|---|---|---|---|---|---|---|---|---|---|---|
| 6 | vBWML17 / hmc | 0.801 | 0.225 | 1.548 | 2 | IL1R / IL1VL | 0.824 | 0.225 | 1.548 | PVT / VD12 | 0.641 | 3.875 | 1.173 |
| 6 | vBWMR13 / hmc | 0.593 | 0.325 | 1.538 | 2 | IL1R / URAVR | 0.872 | 0.225 | 1.548 | RIVL / RIVR | 0.807 | 0.575 | 1.513 |
| 5 | CEPDL / IL1R | 0.716 | 0.050 | 1.566 | 2 | IL1VL / URAVR | 0.632 | 0.225 | 1.548 | VA12 / VD12 | 0.786 | 3.875 | 1.173 |
| 5 | AS11 / vBWMR4 | 0.574 | 1.000 | 1.471 | 2 | ALA / hmc | 0.560 | 3.375 | 1.227 | VB05 / VB06 | 0.950 | 0.050 | 1.566 |
| 5 | VD04 / vBWML5 | 0.617 | 0.075 | 1.563 | 2 | PVR / VD12 | 0.574 | 3.875 | 1.173 | dBWMR1 / GLRDL | 0.884 | 2.000 | 1.369 |
| 5 | vBWML12 / hmc | 0.785 | 0.275 | 1.543 | 2 | DVB / VA12 | 0.590 | 3.875 | 1.173 | dBWMR1 / GLRDR | 0.752 | 2.000 | 1.369 |
| 5 | vBWML18 / hmc | 0.503 | 0.225 | 1.548 | 2 | DVB / VD12 | 0.734 | 3.875 | 1.173 | vBWML5 / hmc | 0.790 | 0.325 | 1.538 |
| 5 | vBWMR12 / hmc | 0.743 | 0.325 | 1.538 | 2 | AVL / VA12 | 0.538 | 3.875 | 1.173 | vBWML7 / hmc | 0.662 | 0.325 | 1.538 |
| 4 | CEPDL / IL1DR | 0.757 | 0.050 | 1.566 | 2 | RMEL / RMED | 0.949 | 2.650 | 1.303 | vBWMR5 / hmc | 0.838 | 0.325 | 1.538 |
| 4 | OLQDL / IL1R | 0.716 | 0.050 | 1.566 | 2 | RMED / dBWMR1 | 0.501 | 2.650 | 1.303 | vBWMR7 / hmc | 0.573 | 0.325 | 1.538 |
| 4 | VD04 / vBWML7 | 0.535 | 0.075 | 1.563 | 2 | DA09 / PDB | 0.592 | 3.000 | 1.266 | dBWMR8 / hmc | 0.946 | 0.125 | 1.558 |
| 4 | dBWML4 / hmc | 0.721 | 0.000 | 1.571 | 2 | PDA / VA12 | 0.586 | 3.875 | 1.173 | vBWML8 / hmc | 0.774 | 0.225 | 1.548 |
| 4 | dBWMR11 / hmc | 0.913 | 0.125 | 1.558 | 2 | PDA / VD12 | 0.766 | 3.875 | 1.173 | CANL / exc_cell | 0.762 | 0.250 | 1.546 |
| 4 | vBWML11 / hmc | 0.705 | 0.275 | 1.543 | 2 | PDB / VD12 | 0.596 | 3.875 | 1.173 | CANR / exc_cell | 0.761 | 0.250 | 1.546 |
| 3 | vBWMR11 / hmc | 0.773 | 0.325 | 1.538 | 2 | VB05 / VB07 | 0.628 | 0.050 | 1.566 | mu_intL / mu_intR | 0.607 | 0.050 | 1.566 |
| 3 | PHCR / VD12 | 0.567 | 3.875 | 1.173 | 2 | vBWML4 / hmc | 0.803 | 0.325 | 1.538 | mu_intL / mu_anal | 0.984 | 0.050 | 1.566 |
| 3 | OLQDL / IL1DR | 0.757 | 0.050 | 1.566 | 2 | vBWML6 / hmc | 0.822 | 0.325 | 1.538 | mu_intL / mu_sph | 0.984 | 0.050 | 1.566 |
| 3 | IL1DR / URAVR | 0.708 | 0.225 | 1.548 | 2 | vBWMR4 / hmc | 0.793 | 0.325 | 1.538 | mu_intR / mu_anal | 0.625 | 0.050 | 1.566 |
| 3 | IL1L / IL1R | 0.859 | 0.225 | 1.548 | 2 | vBWMR6 / hmc | 0.992 | 1.800 | 1.390 | mu_intR / mu_sph | 0.625 | 0.050 | 1.566 |
| 3 | IL1L / URAVR | 0.871 | 0.225 | 1.548 | 2 | dBWMR9 / hmc | 0.807 | 0.125 | 1.558 | vm1AL / vm1PL | 0.573 | 2.025 | 1.367 |
| 3 | ALA / dBWML7 | 0.866 | 0.850 | 1.486 | 2 | vBWML8 / exc_gl | 0.715 | 0.800 | 1.491 | vm1AR / vm1PR | 0.573 | 2.025 | 1.367 |
| 3 | ALA / dBWML8 | 0.867 | 0.850 | 1.486 | 2 | vBWML9 / hmc | 0.707 | 0.225 | 1.548 | vm2AL / vm2AR | 0.927 | 1.700 | 1.400 |
| 3 | ALA / vBWML8 | 0.507 | 0.800 | 1.491 | 2 | GLRL / GLRR | 0.916 | 2.650 | 1.303 | vm1AL / vm1AR | 0.714 | 1.750 | 1.395 |
| 3 | AVFL / ASO3 | 0.625 | 1.000 | 1.471 | 2 | CANL / CANR | 0.999 | 1.950 | 1.375 | vm1AL / vm1PL | 0.795 | 1.750 | 1.395 |
| 3 | dBWML5 / hmc | 0.806 | 0.000 | 1.571 | 2 | mu_anal / mu_sph | 0.984 | 0.050 | 1.566 | vm1AR / vm1PR | 0.795 | 1.750 | 1.395 |
| 3 | dBWMR1 / hmc | 0.869 | 2.000 | 1.369 | 2 | vm2AL / vm2PL | 0.892 | 1.700 | 1.400 | vm1PL / vm1PR | 0.714 | 1.750 | 1.395 |
| 3 | vBWML3 / hmc | 0.604 | 0.325 | 1.538 | 2 | vm2AL / vm2PR | 0.888 | 1.700 | 1.400 | vm2PL / vm2PR | 0.927 | 1.700 | 1.400 |
| 3 | vBWML10 / hmc | 0.797 | 0.225 | 1.548 | 2 | vm2AR / vm2PL | 0.888 | 1.700 | 1.400 | | | | |

