## [Editor Report · eLife Assessment]

This study presents numerical results on a framework for understanding the dynamics of subthreshold waves in a network of electrical synapses modeled on the connectome data of the C elegans nematode. The strength of the evidence presented in favor of interference effects being a major component in subthreshold wave dynamics is **inadequate** and the approach is flawed. Substantial methodological issues are present, including altering the original network structure of the connectome without a clear justification and providing little motivation for the choice of numerical parameters values that were used.

---

## [Referee Report · Reviewer #1 (Public review)]

Summary:

This work investigates numerically the propagation of subthreshold waves in a model neural network that is derived from the *C. elegans* connectome. Using a scattering formalism and tight-binding description of the network -- approximations which are commonplace in condensed matter physics -- this work attempts at showing the relevance of interference phenomena, such as wavenumber-dependent propagation, for the dynamics of subthreshold waves propagating in a network of electrical synapses.

Strengths:

The primary strength of the work is in trying to use theoretical tools from a far-away corner of fundamental physics to shed light on the properties of a real neural system.

Weaknesses:

The authors provide a good introduction and motivation for studying the propagation of subthreshold oscillations in the inferior olive nuclei. However, they chose to use the C elegans connectome for their study, and the implications of this work for C elegans neuroscience remain unclear by the end of the preprint. The authors should also give more evidence for the claim that their study may give a mechanism for synchronized rhythmic activity in the mammalian inferior olive nucleus, or refrain from making this conclusion. In the same vein, since the work emphasizes the dependence on the wavenumber for the propagation of subthreshold oscillations, they should make an attempt at estimating the wavenumber of subthreshold oscillations in C elegans if they were to exist and be observed. Next, the presence of two "mobility edges" in the transmission coefficient calculated in this work is unmistakably due to the discrete nature of the system, coming from the tight-binding approximation, and it is unclear to me if this approximation is justified in the current system. Similarly, it is possible that the wavenumber-dependent transmission observed depends strongly on the addition of a large number of virtual nodes (VNs) in the network, which the authors give little to no motivation for. As these nodes are not present in the C elegans connectome, the authors should explain the motivation for their inclusion in the model and should discuss their consequences on the transmission properties of the network. As it stands, I think the work would only have a very limited impact on the understanding of subthreshold oscillations in the rat or in C elegans. Indeed, the preprint falls short of relating its numerical results to any phenomena which could be observed in the lab.

---

## [Referee Report · Reviewer #2 (Public review)]

This manuscript addresses an interesting and important question: the basic mechanisms underlying subthreshold intrinsic oscillations in the inferior olive. Instead of a direct investigation of the questions, the authors decide to study subthreshold oscillations in the C-elegans, where the connectivity pattern is known but does not exhibit sub-threshold oscillations. Furthermore, instead of the common description of gap-junction coupling by resistors, the authors decide to represent the system as a tight-binding Anderson Hamiltonian.

Weaknesses:

The authors study an architecture of the C-elegans instead of that of the inferior olive of mammals because the architecture of C-elegans is known.

No subthreshold oscillations were identified in the C-elegans.

Instead of representing electrical coupling via resistors that connect neurons, the authors use a quantum formalism and introduce the tight-binding Anderson Hamiltonian. Why?

Equally spaced two virtual nodes were added between cells connected by a gap junction. Why?

Comments on revised version:

Last time, I recommended that the authors should represent electrical coupling via resistors that connect neurons instead of via the quantum formalism. The authors have not tested this direction.

---

## [Author Response]

The following is the authors’ response to the original reviews.

**Joint Public Review:**
(1) This work investigates numerically the propagation of subthreshold waves in a model neural network that is derived from the *C. elegans* connectome. Using a scattering formalism and tight-binding description of the network -- approximations which are commonplace in condensed matter physics -- this work attempts to show the relevance of interference phenomena, such as wavenumber-dependent propagation, for the dynamics of subthreshold waves propagating in a network of electrical synapses.(2) The primary strength of the work is in trying to use theoretical tools from a far-away corner of fundamental physics to shed light on the properties of a real neural system. While a system composed of neurons and synapses is classical in nature, there are occasions in which interference or localization effects are useful for understanding wave propagation in complex media [review, van Rossum & Nieuwenhuizen, 1999]. However, it is expected that localization effects only have an impact in some parameter regimes and with low phase dissipation. The authors should have addressed the existence of this validity regime in detail prior to assuming that interference effects are important.

The theoretical concept and tool used in this study are not situated in a far-away corner of fundamental physics but hold one of the central positions in condensed matter physics and statistical physics. In fact, the non-scientific statement about where the theoretical concept and tool employed by the researchers are positioned within the realm of fundamental physics is irrelevant. The fundamental physics governs the foundations of all natural phenomena, and thus it provides indispensable principles for interpreting not only neural systems but also all life phenomena. One such principle explored in our study is the interference and localization of waves.

Specifically, in the third paragraph of the Introduction, we introduced that the interference effect of subthreshold oscillating waves, beyond being a theoretical possibility, is a phenomenon actually observed in neural tissue (Chiang and Durand, 2023; Gupta et al., 2016). Moreover, according to Devor and Yarom (2002), the propagation of subthreshold oscillations observed in the inferior olivary nucleus extended beyond a distance of 0.2 mm. Therefore, considering the propagation of subthreshold waves and the resulting interference in the connectome of *C. elegans*, which has a total body length of less than 1 mm, a diameter of about 0.08 mm, and most neurons distributed in the ring structure near its neck, provides sufficient validity for the initiation of theoretical and computational studies.

The primary objective of our study is to investigate which regimes of signal transmission/localization and interference phenomena are valid within the network of electrical synapses in *C. elegans*, the only system for which the neural connectome structure is perfectly known. As the Reviewer rightly pointed out in the question, this is exactly the issue that the Reviewer is curious about. Therefore, the existence of this validity regime cannot be addressed prior to conducting the study but can only be identified as a result of performing the research. And we have conducted such a study.

(3) An additional approximation that was made without adequate justification is the use of a tight-binding Hamiltonian. This can be a reasonable approximation, even for classical waves, in particular in the presence of high-quality-factor resonators, where most of the wave amplitude is concentrated on the nodes of the network, and nodes are coupled evanescently with each other. Neither of these conditions were verified for this study.

The tight-binding Anderson Hamiltonian we used in this study originally consisted of the on-site energy at each node and the hopping matrix between nodes. When the on-site energy is relatively much more stable (i.e., has a large negative value) compared to the hopping matrix, most of the wave amplitude becomes concentrated on the nodes as the Reviewer mentioned. However, as is well-known from reference papers (Anderson, 1958; Chang et al., 1995; Meir et al., 1989; Shapir et al., 1982; Thomas and Nakanishi, 2016), in this study, we also removed the on-site energy to prevent the waves from being concentrated on the nodes. Therefore, the tight-binding Hamiltonian we used in this study ensures that waves propagate through edges in the network where the values of the hopping matrix exist.

To assist the Reviewer in better understanding the model used in this study, we provide additional explanations as follows. In the manuscript, we have already provided detailed descriptions of the setup using the tight-binding Anderson Hamiltonian in the Method section under “Construction of our circuit model” and the explanation of Figure 1. In the model we used, the edges represented by solid lines are perfect conductors, while the dotted lines representing gap junctions act as potential barriers (Fig. 1B). Therefore, when electric signals propagate, we are dealing with the phenomenon where signals transmitted through the edges encounter potential barriers, causing scattering or attenuation. The model described by the Reviewer is indeed a commonly used model in condensed matter physics, but we did not use the exact model mentioned by the Reviewer. Instead, as is common in well-known reference papers, we modified it to suit our purposes. We hope this explanation helps the Reviewer gain a better understanding.

(4) The motivation for this work is to understand the basic mechanisms underlying subthreshold intrinsic oscillations in the inferior olive, but detailed connectivity patterns in this brain area are not available. The connectome is known for C elegans, but sub-threshold oscillations have not been observed there, and the implications of this work for C elegans neuroscience remain unclear. The authors should also give more evidence for the claim that their study may give a mechanism for synchronized rhythmic activity in the mammalian inferior olive nucleus, or refrain from making this conclusion.

We agree with the Reviewer's point. In this study, we do not provide additional analysis on the mammalian inferior olive nucleus beyond what is already known from previous research. What we intended to discuss in the Discussion section was to suggest that within our model, there is a “possibility” that a group of cells exchanging wave signals of a specific wavenumber with high transmittance may show synchronized rhythmic activity. Therefore, to avoid any misunderstanding for the reader, we have revised the corresponding sentence in the Discussion as follows.

In the Discussion, “The plausible possibility according to our model study is that the constructive interference of subthreshold membrane potential waves with a specific wavenumber may generate the synchronized rhythmic activation.

(5) In the same vein, since the work emphasizes the dependence on the wavenumber for the propagation of subthreshold oscillations, they should make an attempt at estimating the wavenumber of subthreshold oscillations in C elegans if they were to exist and be observed. Next, the presence of two "mobility edges" in the transmission coefficient calculated in this work is unmistakably due to the discrete nature of the system, coming from the tight-binding approximation, and it is unclear if this approximation is justified in the current system.

In this study, we modeled the propagation of subthreshold waves on the electrical synapse network of *C. elegans*, but we did not explain the generation of subthreshold oscillations themselves. Here, we simply injected wave signals with various wavenumber values into the network using a hypothetical device called an "Injector." As the Reviewer pointed out, estimating the wavenumbers of subthreshold oscillations that may exist or be observed in *C. elegans* would require a comprehensive investigation of the membrane potential dynamics occurring in the membranes of individual neurons. However, this is beyond the scope of this study and would require considerable effort to accomplish.

As for the use of the tight-binding Hamiltonian, we have addressed that in our response to the third paragraph in the Joint Public Review above.

(6) Similarly, it is possible that the wavenumber-dependent transmission observed depends strongly on the addition of a large number of virtual nodes (VNs) in the network, which the authors give little to no motivation for. As these nodes are not present in the C elegans connectome, the authors should explain the motivation for their inclusion in the model and should discuss their consequences on the transmission properties of the network.

As mentioned in our response to the third paragraph in the Joint Public Review above, in our model, a node is simply a pathway for waves to pass through. Therefore, inserting virtual nodes between two neurons that are connected in the *C. elegans* connectome does not alter the actual connection structure. In other words, virtual nodes do not create new connections between cells that didn’t exist in the connectome. The virtual nodes we introduced are merely a way to divide the sections—axon, gap junction, dendrite—through which the wave passes when it is transmitted between two neurons. As we have already explained in Fig. 1B, the edge connected by two virtual nodes, represented by a dotted line, is motivated to depict the gap junction acting as a potential barrier. We hope this explanation helps the Reviewer better understand the model used in this study.

(7) As it stands, the work would only have a very limited impact on the understanding of subthreshold oscillations in the rat or in C elegans. Indeed, the preprint falls short of relating its numerical results to any phenomena which could be observed in the lab.

In this study, we proposed a minimalistic model built using the currently available but limited *C. elegans* connectome information. Specifically, our model is not a phenomenological one that adjusts parameters to accurately predict experimental measurements, but rather an attempt at a novel conceptual approach to theoretically possible scenarios. While the model may not be satisfactory enough to explain experimental phenomena at present, it is a theoretical/computational study that someone needs to undertake. We believe this is the path of scientific progress. Therefore, as the Reviewer has expressed concern, it is entirely understandable that reproducing the numerical results measured in actual experiments is difficult in this study. Nevertheless, we believe that this study makes a basic contribution to the conceptual understanding of subthreshold signal propagation in *C. elegans’* electric synapses.

Rather than offering a stretched opinion, we maintain a positive hope that future researchers in this field will improve the model by incorporating more detailed and extensive biological data through follow-up studies, allowing us to get closer to describing real phenomena.

**Recommendations for the authors:**

**Reviewer #1 (Recommendations for the authors):**
The word "Sensory" was misspelled in Figures 2, 4 and 5.

We appreciate the feedback from Reviewer #1. We have corrected the mentioned typos in Figures 2, 4, and 5 of the revised manuscript.